# Estrogen-related receptor gamma functions as a tumor suppressor in gastric cancer

Myoung-Hee Kang [1,2,3], Hyunji Choi[4], Masanobu Oshima[5], Jae-Ho Cheong[6], Seokho Kim[7], Jung Hoon Lee [8], Young Soo Park[9], Hueng-Sik Choi[10], Mi-Na Kweon[2], Chan-Gi Pack[1,2], Ju-Seog Lee [3], Gordon B. Mills[3], Seung-Jae Myung[1,2,8] & Yun-Yong Park[1,2]

The principle factors underlying gastric cancer (GC) development and outcomes are not well characterized resulting in a paucity of validated therapeutic targets. To identify potential molecular targets, we analyze gene expression data from GC patients and identify the nuclear receptor ESRRG as a candidate tumor suppressor. ESRRG expression is decreased in GC and is a predictor of a poor clinical outcome. Importantly, ESRRG suppresses GC cell growth and tumorigenesis. Gene expression profiling suggests that ESRRG antagonizes Wnt signaling via the suppression of TCF4/LEF1 binding to the *CCND1* promoter. Indeed, ESRRG levels are found to be inversely correlated with Wnt signaling-associated genes in GC patients. Strikingly, the ESRRG agonist DY131 suppresses cancer growth and represses the expression of Wnt signaling genes. Our present findings thus demonstrate that ESRRG functions as a negative regulator of the Wnt signaling pathway in GC and is a potential therapeutic target for this cancer.

[1] ASAN Institute for Life Sciences, ASAN Medical Center, University of Ulsan College of Medicine, Seoul 05505, Republic of Korea. [2] Department of Convergence Medicine, University of Ulsan College of Medicine, Seoul 05505, Republic of Korea. [3] Department of Systems Biology, MD Anderson Cancer Center, Houston, TX 77030, USA. [4] Department of Biological Sciences, Dong-A University, Busan 49315, Republic of Korea. [5] Division of Genetics, Cancer Research Institute, Kanazawa University, Kanazawa 920-8641, Japan. [6] Department of Surgery, Yonsei University College of Medicine, Seoul 03722, Republic of Korea. [7] Aging Research Institute, Korea Research Institute of Bioscience and Biotechnology, Daejeon 34141, Republic of Korea. [8] Department of Gastroenterology, University of Ulsan College of Medicine, Seoul 05505, Republic of Korea. [9] Department of Pathology, University of Ulsan College of Medicine, Seoul 05505, Republic of Korea. [10] National Creative Research Initiatives Center for Nuclear Receptor Signals and Hormone Research Center, School of Biological Sciences and Technology, Chonnam National University, Gwangju 61186, Republic of Korea. Correspondence and requests for materials should be addressed to S.-J.M. (email: sjmyung@amc.seoul.kr) or to Y.-Y.P. (email: yypark@amc.seoul.kr)

Gastrointestinal (GI) cancers are among the most common cancers worldwide[1]. Among GI cancers, gastric cancer (GC) is the predominant cause of mortality in Asian populations[1]. Surgical resection in the case of stage I and II GC patients and adjuvant chemotherapy are currently the standard treatments for GC[2]. Recently, molecular therapeutics have been implemented to target GC. These include trastuzumab, which targets HER2, and bevacizumab, which targets VEGF-A[3]. Unfortunately only 5–10% of GC cases are HER2-positive, and not all of the patients in this subset respond to trastuzumab, demonstrating the urgent need to identify new molecular targets to impact GC patient outcomes[4,5].

The biological complexity of GC has hampered the discovery of molecular targets and subsequent implementation of targeted therapies[6]. Thus, a better understanding of the molecular drivers of GC pathophysiology is essential for the identification of novel therapeutic targets[3,6]. An imbalance between tumor suppressors and oncogenes influences cancer development across multiple tumor lineages[7]. TP53, PTEN, and RUNX3 have been implicated as tumor suppressors in GC[8,9]. Although the molecular mechanisms of tumor suppression are diverse, deregulation of any of these factors is a critical step in tumorigenesis[10]. TP53 and RUNX3 function as transcription factors (TFs) and confer tumor suppressive activity by antagonizing diverse oncogenic pathways including the Wnt and TGF-β pathways. Thus, key TFs are well-recognized tumor suppressors.

In our current study, we identified estrogen-related receptor gamma (ESRRG; also known as ERRγ) as a potential tumor suppressor in GC by genomic analysis. ESRRG and its specific agonist, DY131, were found to inhibit GC cell growth, and patients harboring ESRRG gene signatures showed an improved prognosis. In addition, genomic profiling analysis revealed that, similar to other tumor suppressor genes in GC, ESRRG suppresses the Wnt signaling pathway. Our present study thus provides new insights into the molecular mechanisms in GC, and suggests that activation of ESRRG by antagonizing Wnt signaling through compounds such as DY131 could provide a novel therapeutic approach to treating this cancer.

## Results

**Identification of ESRRG as a tumor suppressor in GC.** Recently, genomic data analysis has been used to uncover previously unknown functions of various genes involved in cancer[7,11]. We carried out genomic analysis of publicly available gene expression data (GSE13861[6], GSE26899, GSE29272). To screen for genes differentially expressed in GC, we compared normal gastric samples to tumor samples by applying class comparison analysis[12]. We identified 521 genes as being potentially cancer-associated (Fig. 1a). Of these genes, we focused on TFs for further analysis as they are the regulatory endpoints of signaling pathways and their deregulation is commonly linked to cancer development[7]. Among the TFs in this gene panel, we selected those that could be potential drug targets. Since nuclear receptors (NRs) possess a ligand-binding pocket[13], we hypothesized that they would be good candidates in this respect. When genes were ranked according to fold changes between GC and normal gastric samples, ESRRG was one of top-ranked TFs and NRs, exhibiting a greater than 10-fold downregulation in cancer tissues (normal vs. tumor: -14.851 fold in GSE29272; -16.514 fold in GSE26899; -23.608 fold in GSE13861; Fig. 1b, Supplementary Fig. 1a and b). These results were validated in independent cohorts using western blotting and quantitative real-time reverse transcriptase PCR (qRT-PCR) (Fig. 1c, d and Supplementary Fig. 1c). We then focused on elucidating the function of ESRRG in GC. ESRRG is a member of the ESRR nuclear receptor family[14], which also

includes ESRRA and ESRRB that were found to be predominantly expressed in normal gastric tissues (Supplementary Fig. 2a–d).

To determine whether ESRRG expression is lost during gastric oncogenesis, we took advantage of the *Gan*-mouse model system where an initially induced gastritis sometimes progresses to malignant GC[15]. Indeed, ESRRG expression was found to be significantly suppressed in the *Gan* mice in gastric hyperplasia and decreased further in dysplastic tumors (Fig. 1e and f). These results suggested that ESRRG expression is lost during GC development, either as a consequence of oncogenic transformation or due to its role as a tumor suppressor.

**ESRRG suppresses GC cell growth.** To investigate whether ESRRG functions as a tumor suppressor, we examined whether it affects cancer cell growth. A panel of GC cell lines including AGS, NCI-N87, MKN45, and MKN28 were stably infected with ESRRG-expressing lentivirus vectors (Supplementary Fig. 3a and b). In all cell lines assayed, ESRRG overexpression led to a significant inhibition of monolayer cell growth and colony formation (Fig. 2a, b and Supplementary Fig. 4a–c). To assess the impact of ESRRG on in vivo tumor growth, NCI-N87 cells with or without ectopic overexpression of ESRRG were subcutaneously transplanted into athymic nude mice and tumor growth was monitored. As expected, ESRRG suppressed tumor growth in this in vivo mouse model (Fig. 2c). In addition, tumor volume and weight were also significantly decreased in ESRRG-overexpressing NCI-N87 cells (Fig. 2d). Consistent with the decreases observed in tumor growth, we observed reduced cell proliferation, as assessed by Ki67 expression, upon ESRRG overexpression (Fig. 2e and f). Our in vitro and in vivo experiments thus collectively demonstrated that ESRRG plays a tumor suppressive role in GC.

**Clinical relevance of ESRRG in GC patients.** The aberrant expression of TFs has frequently been found to dictate the clinical outcome in cancer patients[7]. We evaluated whether ESRRG itself has clinical relevance in GC. Patient cohorts from GEO (Gene Expression Omnibus in the National Center for Biotechnology Information) were dichotomized according to ESRRG expression. Patients with a higher ESRRG level had good clinical outcomes and vice versa across multiple sample sets (Fig. 3a), consistent with a tumor suppressor role of ESRRG in GC.

To investigate potential downstream targets that mediate the functions of ESRRG, we undertook gene expression profiling and detected 3009 genes that are differentially regulated between control and ESRRG OE (overexpression) groups in AGS cells. We selected 435 genes exhibiting a greater than 1.75-fold change between control and ESRRG OE (Fig. 3b). We subsequently examined the clinical relevance of the ESRRG activity using its gene signature and a previously established prediction strategy that employs multiple different algorithms[11]. Interestingly, patients with ESRRG OE signatures showed significantly better overall survival (OS) and relapse free survival (RFS) outcomes compared with ESRRG Con. signatures (Fig. 3c and Supplementary Fig. 5). This genomic analysis of GC patients strongly suggested that ESRRG is highly associated with the prognosis in GC and could be a powerful indicator of clinical outcome in these cases.

**ESRRG antagonizes the Wnt pathway in GC.** We investigated the mechanisms underlying the ESRRG-mediated suppression of tumor growth. Our data showed that the downregulated genes in ESRRG-overexpressing cells included oncogenic factors such as *CCND1*, *PCNA*, *TOP2B*, *SKP1*, *JAG1*, and multiple genes involved in the Wnt signaling pathway (Fig. 4a). Since Wnt signaling contributes to oncogenic potential and it is an attractive

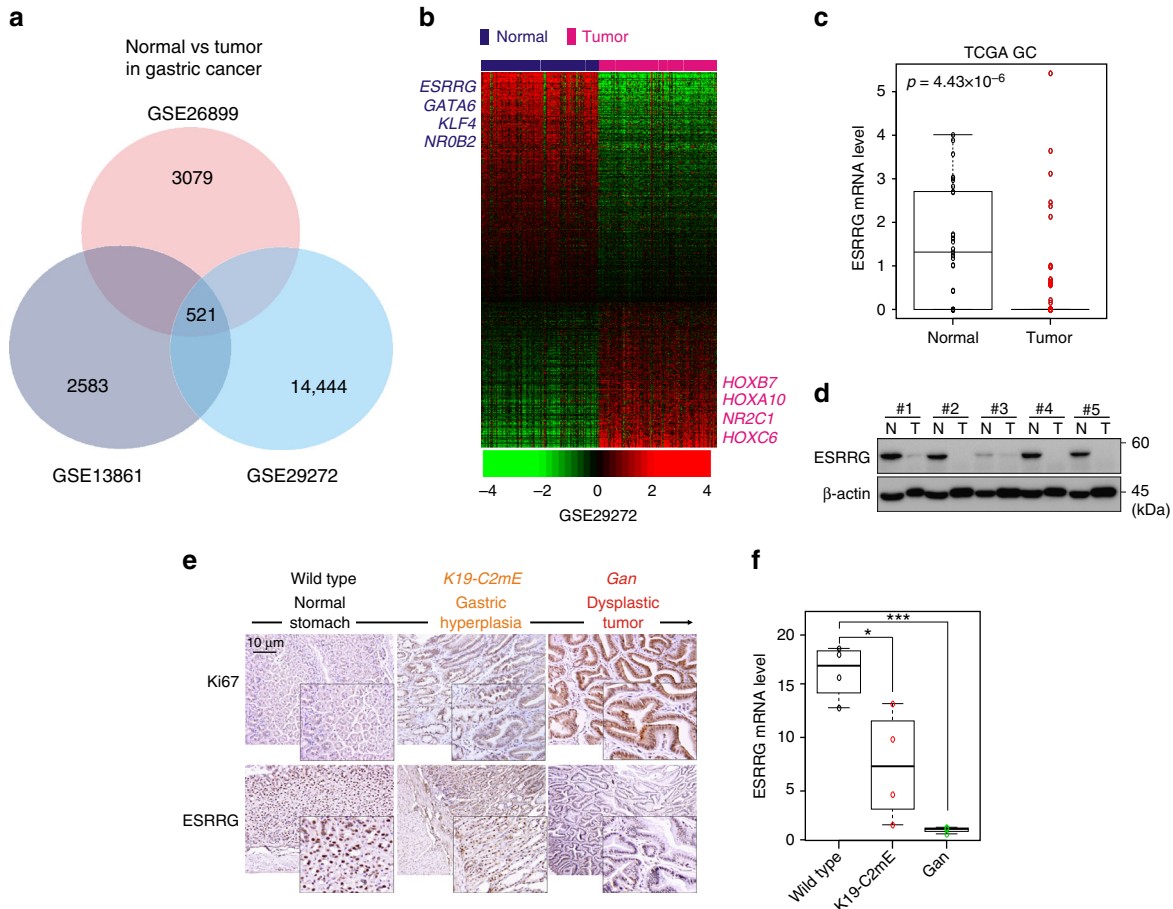

**Fig. 1** ESRRG expression in GC patients and in a mouse model. **a** Venn diagram of genes showing significant differential expression between normal and cancer tissue in the three different GC patient cohorts. A univariate test (two-sample $t$-test) with a multivariate permutation test (10,000 random permutations) was employed. In each comparison, a cut-off $p$-value of less than 0.001 was applied to retain genes with an expression level that differed significantly between the two groups of tissues examined. **b** Expression patterns of selected genes shared in the three GC patient cohorts. The expression of 521 genes was commonly up- or downregulated in all three cohorts. Colored bars at the top of the heat map represent samples as indicated. Genes involved in transcription are highlighted in blue or red text. **c** ESRRG expression from TCGA data. **d** Western blot analysis from normal gastric and tumor tissues. **e**, **f** Gastritis in K19-C2mE mice and gastric tumors in K19-Wnt1/C2mE mice. **e** Immunohistochemistry (IHC) staining of Ki67 and ESRRG in wild type (left), K19-C2mE (middle), and Gan (right) mouse stomachs. **f** ESRRG mRNA levels in the GC mouse model. Original magnification, ×200. Data represent the mean ± s.d. (error bars) from the indicated samples ($n = 4$ per group). Student's $t$-test was used to examine statistical significance (* $p < 0.05$, *** $p < 0.005$)

therapeutic target being currently explored for cancer therapy[16], we investigated the role of ESRRG in regulating the Wnt pathway. First, qRT-PCR was performed to validate the gene expression profiling data, and revealed that Wnt-associated genes such as *DVL3*, *LEF1*, *LGR5*, *TCF7L2*, *AXIN2*, and *CTNNB1* were significantly downregulated in AGS and MKN28 GC cells following ESRRG transfection (Fig. 4b). We next evaluated whether the tumor suppressive properties of ESRRG were due to the suppression of Wnt signaling. Whereas cell growth was reduced by ESRRG overexpression, the ectopic expression of the TCF4/TCF7L2 and LEF1 Wnt effector genes accelerated cell growth in cells with ESRRG overexpression, suggesting that ESRRG-induced growth suppression is reversed by TCF4 and LEF1 (Fig. 4c, d and Supplementary Fig. 6a). Consistently, inhibition of tumor growth by ESRRG overexpression in a xenograft mouse model was rescued by the re-introduction of TCF4/LEF1, and Ki67 expression was also rescued (Fig. 4e and f).

We examined the effect of ESRRG on Wnt target gene activity. As shown previously, the constitutively active β-catenin mutant, CTNNB1 (also known as β-catenin) S37A, markedly enhanced the transcriptional activity of the Top/Flash reporter, which has

multiple binding sites for TCF/LEF. This transactivation was significantly repressed by ESRRG (Fig. 4g), however, as was the increased *CCND1* promoter activity induced by TCF4/LEF1 (Fig. 4h and Supplementary Fig. 6b). In contrast, ESRRG could not directly activate or suppress *CCND1* promoter activity. Our data indicated, therefore, that ESRRG likely functions as a transcriptional repressor of Wnt target genes by indirect mechanisms.

We also investigated the clinical relevance of ESRRG and Wnt signaling associated genes using GC patient samples. Wnt signaling associated genes such as *LEF1*, *TCF4/TCF7L2*, *AXIN2*, *CTNNB1*, *DVL3*, and *LGR5* were found to be expressed at markedly higher levels in GC samples compared with normal tissues (Supplementary Fig. 7), which was inversely correlated with ESRRG mRNA expression (Fig. 4i and Supplementary Fig. 8). Kaplan–Meier analyses of dichotomized gene expression showed that a higher expression of those genes was associated with significantly poorer clinical outcomes (Supplementary Fig. 9). Our data thus suggest that ESRRG negatively regulates Wnt signaling components and that this contributes to its tumor suppressive properties.

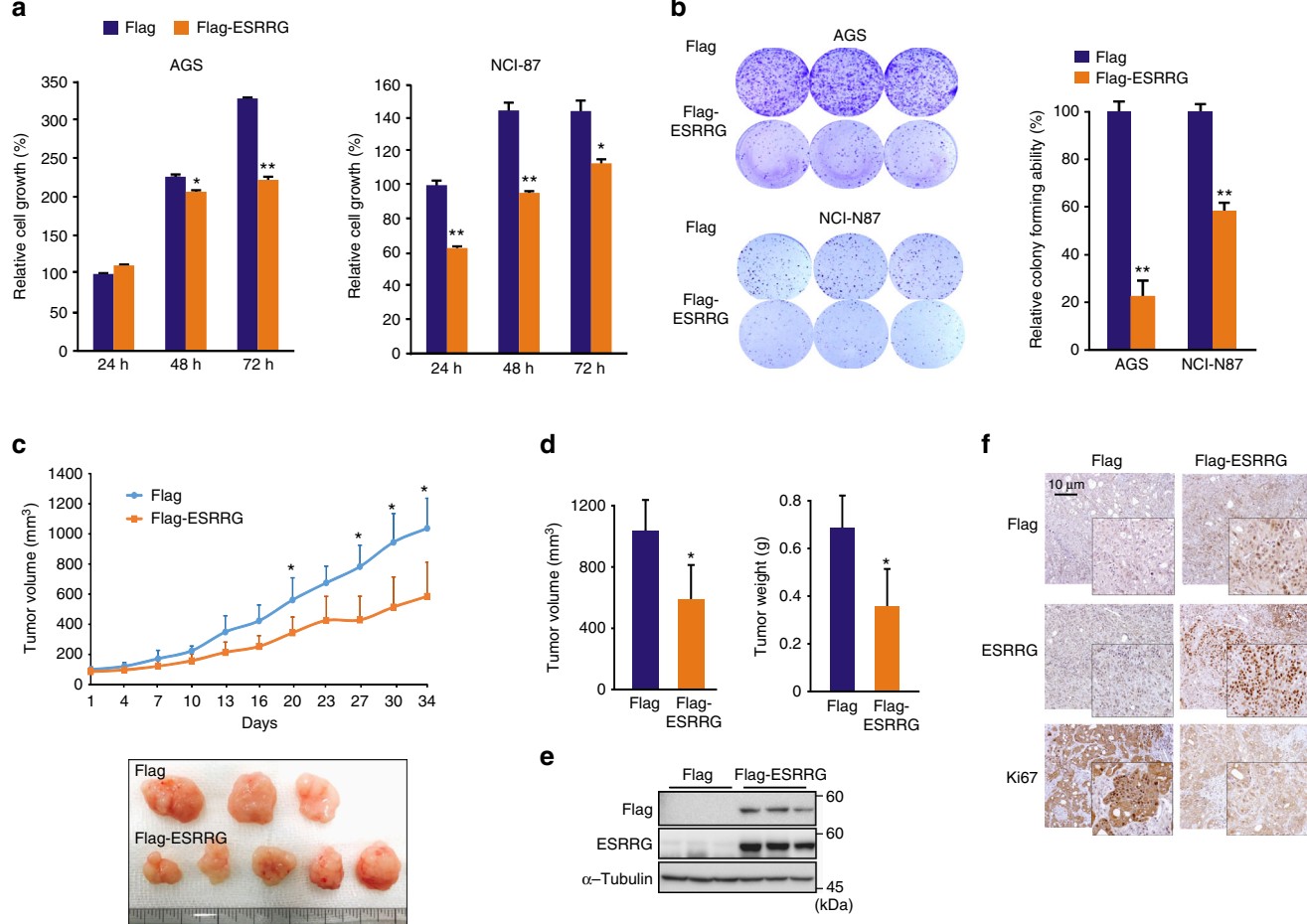

**Fig. 2** Effect of ESRRG on GC cell growth. **a** CCK8 assay and **b** colony formation assay after ESRRG overexpression in AGS and NCI-N87 cells ($n = 3$). **c–f** In vivo GC tumorigenesis analysis in xenograft nude mouse models (Flag: $n = 6$; Flag-ESRRG: $n = 7$). **c** ESRRG overexpression inhibited tumor growth in the xenograft nude mouse model. Flag-control, NCI-N87 cells, or an NCI-N87 cell line infected with a Flag-ESRRG overexpression vector (ESRRG-OE), were injected into nude mice and the tumor volume was measured at the indicated time points. **d** Mice were sacrificed and tumor volumes and weights were measured. Western blot (**e**) and IHC (**f**) analysis of mouse samples was then performed. Data represent the mean ± s.d. from the three independent replicates. Student's $t$-test was used to examine statistical significance (*$p < 0.05$, **$p < 0.01$, ***$p < 0.005$)

**Antagonism of the Wnt pathway by ESRRG in GC**. We wished to better understand mechanistically how ESRRG represses Wnt signaling via β-catenin and TCF4/LEF1, which are downstream effector molecules known to confer oncogenic potential in cancer[17,18]. Phosphorylated-β-catenin is its inactive form in canonical Wnt signaling. The active non-phosphorylated β-catenin increases the binding affinity of TCF4/LEF1 to target genes. Thus, we measured the phosphorylation level of β-catenin using western blotting and ELISA. When ESRRG was overexpressed in GC cells, the total β-catenin protein level was unaltered. However, the phosphorylated β-catenin level in ESRRG-overexpressing cells was increased (Fig. 5a). These results were validated by ELISA analysis (Supplementary Fig. 10a), which suggested that ESRRG influences Wnt signaling activity by modulating the β-catenin phosphorylation status. We additionally found that β-catenin degradation by ESRRG is not dependent on ubiquitination or altered GSK3α/β activity (Supplementary Fig. 10b). Activated β-catenin is localized in the nucleus where it forms a complex with TCF/LEF to increase transcriptional activity[17,18]. Hence, we investigated whether the ESRRG regulation of gene expression is dependent on the cellular fraction. Interestingly, our results showed that ESRRG suppressed β-catenin, TCF4, and LEF1 expression in the nucleus but not in the cytoplasm (Fig. 5b). In

addition, the phosphorylated β-catenin level in the cytoplasmic fraction was increased by ESRRG overexpression. Since ESRRG inhibits the expression of Wnt components, we hypothesized that it could influence the stability of these factors. After treatment with the protein synthesis inhibitor cycloheximide (CHX), we measured the expression of Wnt components in cells overexpressing ESRRG. As shown in Fig. 5c, Wnt pathway components were more rapidly degraded in ESRRG-overexpressing cells treated with CHX. In addition, the protein level of Wnt components in ESRRG-overexpressing cells in response to CHX was decreased in the nuclear fraction but not in the cytoplasm (Fig. 5d). This suggested that ESRRG influences Wnt component stability in the manner of a nuclear TF.

The Wnt pathway effector LEF4/TCF1 binds directly to the CCND1 promoter region as a TF[18]. We hypothesized that ESRRG may interfere with this process. Indeed, chromatin immunoprecipitation (ChIP) assays revealed that TCF4/LEF1 binding to its consensus sequence within the CCND1 promoter is blocked by ESRRG (Fig. 5e). In addition, using ChIP analysis with an ESRRG antibody, we observed that CCND1 promoter bound with TCF4/LEF1 was recruited by ESRRG, thus suggesting a direct interaction (Fig. 5f). We next examined whether ESRRG directly interacted with β-catenin or TCF4/LEF1 using IP analysis. The

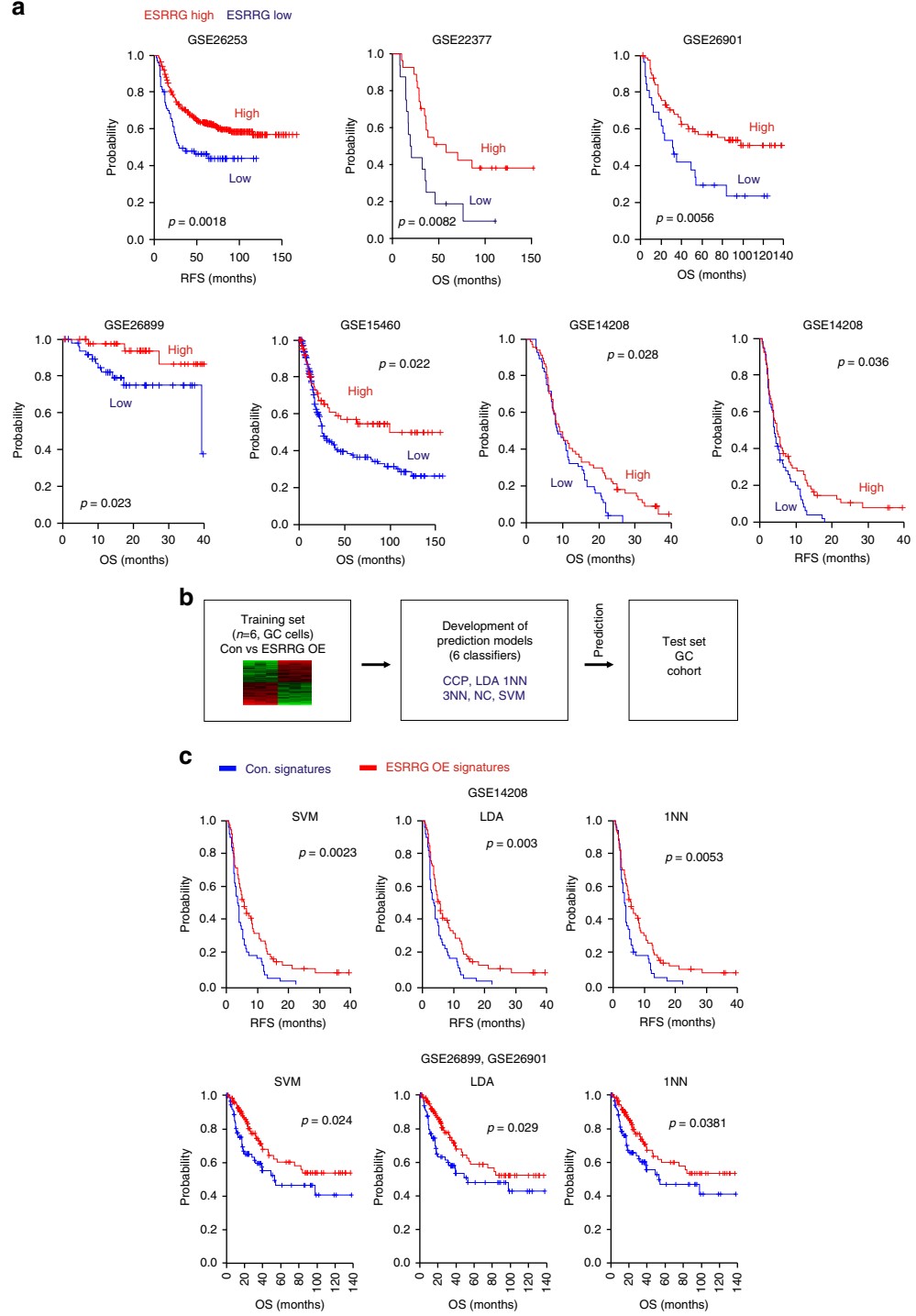

**Fig. 3** Clinical relevance of ESRRG in GC patient prognosis. **a** Patients in the indicated GC cohorts were dichotomized by relatively high or relatively low ESSRG expression and were considered for plotting. **b**, **c** Establishment of gene expression profiles of ESRRG downstream targets. **b** Gene expression signature of ESRRG in control (Con.) or ESRRG lentivirus transduced AGS cells. Genes in the Venn diagram were selected by applying a two-sample Student's *t*-test (*p* < 0.001) to compare Con. and ESRRG OE samples. **b** Overall scheme of prediction models and evaluation of predicted outcome based on a shared gene expression signature of ESRRG in GC cell lines. The ESRRG gene expression signature was used to form a series of classifiers that estimated the probability of how much the expression pattern of a particular patient with GC was similar to the shared signature; control (Con.) vs. ESRRG OE (over-expression). **c** Kaplan–Meier plots of OS of GC patients in the GSE26899 cohort merged with the GSE26901 and GSE14208 cohorts were used to predict outcome based on the Con. and ESRRG OE signatures as a classifier. The differences between groups were significant where indicated (log-rank test). SVM, support vector machines; LDA, linear discriminator analysis; 1NN, one nearest neighbors

results revealed that ESRRG directly interacts with TCF4/LEF1 but not with β-catenin (Fig. 5g). We also examined the co-localization of ESRRG and TCF4/LEF1. As shown in Figs. 5h and 5i, ESRRG indeed co-localized with TCF4/LEF1. Additionally, we investigated interactions among these proteins when tagged with GFP or mCherry in live cells using dual-color fluorescence cross-correlation spectroscopy (FCCS), a highly sensitive method for determining the mobility and interaction of probed molecules[19,20]. We performed dual-color FCCS analysis in live cells

that co-expressed GFP-ESRRG and mCherry-TCF4 or mCherry-LEF1. If the two proteins form a complex, the co-diffusion of GFP and mCherry tagged proteins is detectable as they transit the detection volume. The strength of the interaction (i.e., co-diffusion) is represented by the relative cross-correlation amplitude (see also Methods). A significant interaction was detected between GFP-ESRRG and mCherry-TCF4 or LEF1 compared with the corresponding GFP and mCherry monomers (Fig. 5j and k). We also observed the co-localization of TCF4/LEF

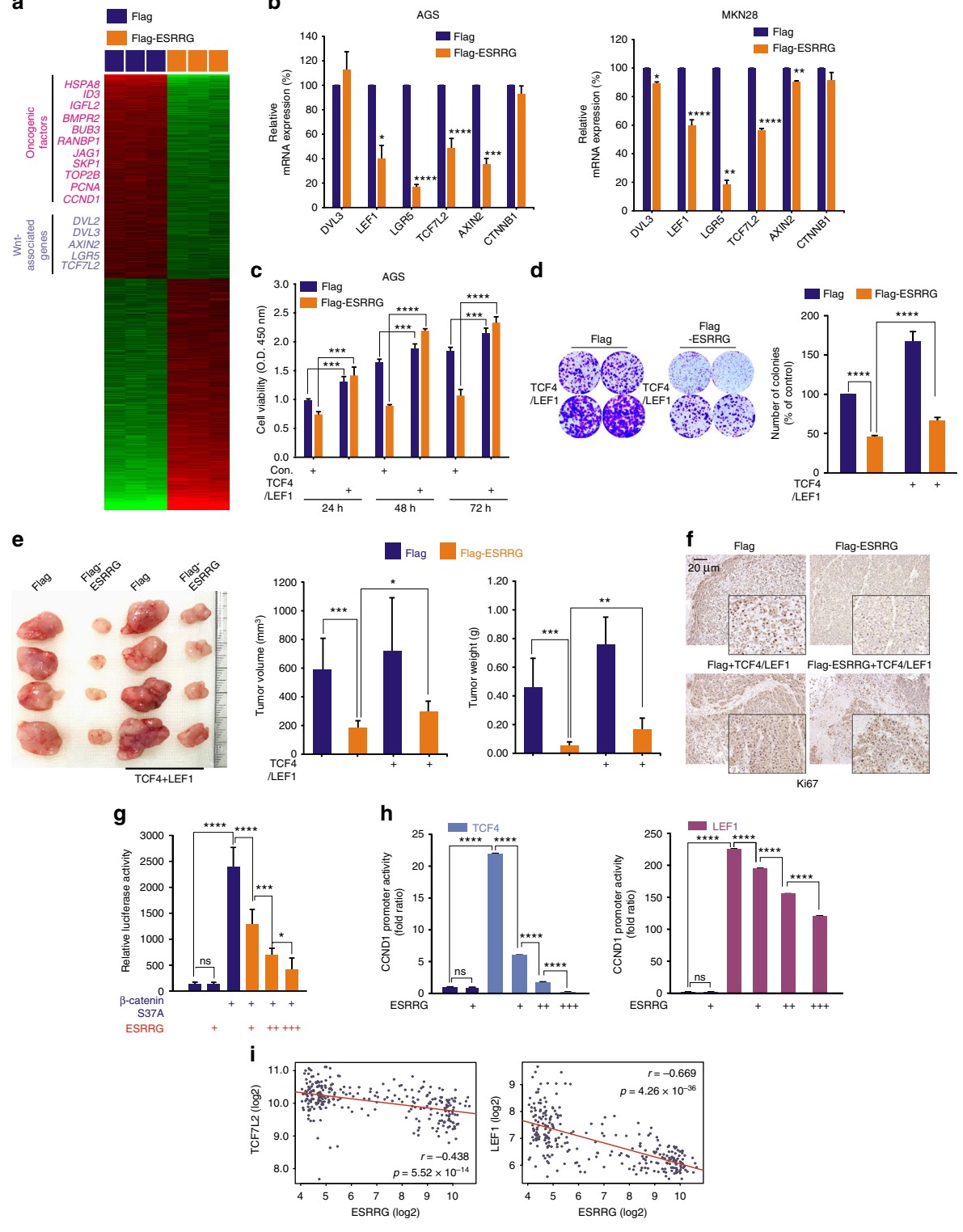

and ESRRG using immunofluorescence in a GC patient-derived organoid model (Fig. 5l).

Upon Wnt activation, β-catenin translocation from the cytoplasm to the nucleus is crucial to activate Wnt-target genes. To test whether ESRRG influences β-catenin translocation, we performed immunofluorescence analysis of GC cells and found that β-catenin was predominantly expressed in the nucleus and that ESRRG overexpression had no effect on its nuclear localization (Supplementary Fig. 10c).

We conclude from these findings that ESRRG, which functions as a transcriptional repressor, antagonizes Wnt signaling by suppressing TCF4/LEF1 binding to the *CCND1* promoter.

**Therapeutic efficacy of the ESRRG agonist DY131 in GC.** Since we found that ESRRG efficiently inhibited cancer cell growth by antagonizing Wnt signaling, we speculated that a pharmacological ESRRG agonist, DY131, might have efficacy as a suppressor of GC. We thus exposed the GC cell types, AGS, MKN28, and NCI-N87, to this agent to examine its possible anti-tumor effects. Indeed, GC cell growth and colony formation were significantly reduced following DY131 treatment (Fig. 6a and Supplementary Fig. 11). DY131 appeared to mediate its effects through ESRRG as it was without activity in ESRRG-silenced cells (Supplementary Fig. 12). DY131 suppressed the expression of Wnt signaling-associated genes (Supplementary Fig. 13a). In addition, the suppression of cell growth by DY131 was rescued by a re-introduction of TCF4/LEF1 (Fig. 6b, c and Supplementary Fig. 13b), confirming that Wnt signaling is directly antagonized by ESRRG and that the ESRRG activator DY131 also influences this pathway. We also treated xenograft tumors with DY131. Consistent with the aforementioned in vitro observations, both tumor volumes and weights were significantly reduced upon treatment with DY131 (Figs. 6d, 6e and Supplementary Fig. 14). Further DY131 treatment of xenograft tumor samples led to decreased expression of Ki67, the three Wnt components (TCF4, LEF1, and β-catenin), and the CCND1 Wnt downstream target gene (Fig. 6f). We also examined the growth inhibitory effects of DY131 in GC patient-derived organoids. Consistently, DY131 treatment significantly inhibited organoid growth and suppressed the expression of Wnt signaling associated genes (Fig. 6g, h and i). Furthermore, reduced organoid proliferation by DY131 was rescued by the re-introduction of TCF4/LEF1 (Fig. 6j and k). Compared with Wnt inhibitors (XAV-939, ICG-001, and Wnt C59), DY131 showed more potency in GC cells (Supplementary Figs. 11 and 15), indicating that Wnt antagonism via ESRRG is effective in inhibiting GC cell growth.

These results clearly demonstrated that the ESRRG agonist DY131 exhibits anti-tumor activity in GC cells by suppressing Wnt signaling.

## Discussion

We have identified a novel tumor-suppressive role for ESRRG in GC that is mediated via the antagonism of Wnt signaling. ESRRG is a member of a NR superfamily of TFs[14] and is specifically expressed in normal stomach and brain[21,22]. Previous reports have implicated ESRRG in the pathophysiology of human breast, endometrial, and prostate cancer[23–25], but a detailed understanding of how ESRRG contributes to cancer progression is still lacking. ESRRG has been proposed previously to function as a tumor suppressor in prostate cancer by arresting the cell cycle via the induction of p21[WAF1/CIP1] and p27[KIP1] [25]. In liver cancer, however, ESRRG appears to exert oncogenic potential by suppressing p21 and p27[26]. Thus, the effects of ESRRG are likely to be cell context-dependent.

ESRRG alters the expression of a plethora of genes that could potentially contribute to its effects in GC. Since these genes include multiple members of the Wnt signaling pathway, we focused on this mechanism. Indeed, our data demonstrate that the Wnt signaling-associated genes *DVL3*, *LEF1*, *LGR5*, *TCF7L2*/*TCF4*, *AXIN2*, and *CTNNB1*, were significantly downregulated by ESRRG in both GC cells and GC patient-derived organoid models (Figs. 4 and 6). Mechanistically, our data clearly demonstrate that ESRRG directly interacts with TCF4/LEF1, which are major TFs governing tumorigenesis, and disrupts functional TCF4/LEF1 binding to the *CCND1* gene promoter region (Figs. 5 and 7). This indicates that ESRRG could modulate the transcriptional activity of Wnt signaling by regulating TCF4/LEF1 gene expression and β-catenin activity (Fig. 7). Using biochemical and advanced techniques such as FCCS, we could clearly demonstrate that ESRRG directly interacts with TCF4/LEF1 and prevents its binding to the *CCND1* promoter (Fig. 5). Transcriptional factors frequently suppress the DNA binding affinity of other molecules to modulate target gene expression[27].

Our present data also indicate that ESRRG influences β-catenin phosphorylation in the cytoplasm (Fig. 5b–d). However, since ESRRG is principally expressed in the nucleus and does not interact with β-catenin, this effect is likely to be an indirect effect potentially due to decreased cell proliferation. As shown in Fig. 5b, whilst ESRRG suppresses β-catenin expression in the nucleus, it does not downregulate β-catenin expression in the cytoplasm. Since ESRRG does not directly interact with β-catenin (Fig. 5g), it may affect nuclear β-catenin expression via an indirect pathway. Regulation of TCF4/LEF1 is directly controlled by binding to and activating a consensus LEF/LEF binding site within its own promoter. ESRRG inhibits transcription activity of TCF4/LEF1 and finally suppresses gene expression level of TCF4/LEF1. Mechanistically, TCF4/LEF1 forms a complex with β-catenin to bind its target gene promoters and promote cancer cell proliferation[28]. Since ESRRG directly interacts with TCF4/LEF1 and inhibits gene transcription and expression, β-catenin might be indirectly suppressed by ESRRG via TCF4/LEF1 inhibition. Although our current findings suggest that ESRRG also influences

---

**Fig. 4** ESRRG antagonizes Wnt signaling in GC. **a** Gene expression profile presented in a matrix format; each row represents an individual gene, and each column represents a transfected cell condition. In this matrix, red and green reflect relatively high and low expression, respectively, as indicated in the scale bar (log₂-transformed scale). Genes associated with oncogenic potential and Wnt signaling-associated genes are listed. **b** qPCR analysis of Wnt signaling associated genes in GC cells (AGS and MKN28) after infection with Flag or Flag-ESRRG lentivirus. **c**, **d** Rescue experiments following the introduction of TCF4/LEF1. After infection of Flag or Flag-ESRRG, the indicated plasmids were transfected into AGS cells, and CCK8 and CFA assays were done. **e**, **f** The MKN45 cell line infected with Flag-ESRRG overexpression vector (ESRRG-OE) or Flag with TCF4/LEF1 was injected into female nude mice and the tumor volume was measured at the indicated time points (*n* = 5 per group). **e** Mice were sacrificed and tumor volumes and weights were measured. **f** IHC analysis from the mouse samples was then performed. **g**, **h** AGS cells were transiently transfected as indicated with the Top/Flash reporter (**g**) or CCND1 promoter (**h**) and the indicated constructs and reporter activity was measured using a luminometer. **i** Correlation of ESRRG and Wnt component gene expression (TCF7L2/TCF4, LEF1) in GSE29272 GC patient cohorts. Scatter plots of ESRRG and Wnt signaling genes in the GC cohorts are shown. Data represent the mean ± s.d. from three independent replicates. Student's *t*-test was used to examine statistical significance (**p* < 0.05, ***p* < 0.01, ****p* < 0.005, *****p* < 0.001)

β-catenin activity, our analysis has clearly demonstrated that ESRRG antagonizes Wnt-signaling as a transcriptional repressor of TCF4/LEF1. Further studies are needed to more fully elucidate how ESRRG influences the Wnt signaling pathway via diverse Wnt components.

It has been difficult to directly target Wnt signaling owing to the lack of pathway-specific targets and the potential redundancy of many pathway components[16]. Since our present data suggest that ESRRG functions as a negative transcriptional regulator of Wnt signaling, it was logical to suppose that agonists of ESRRG activity such as DY131 could also have the potential to efficiently

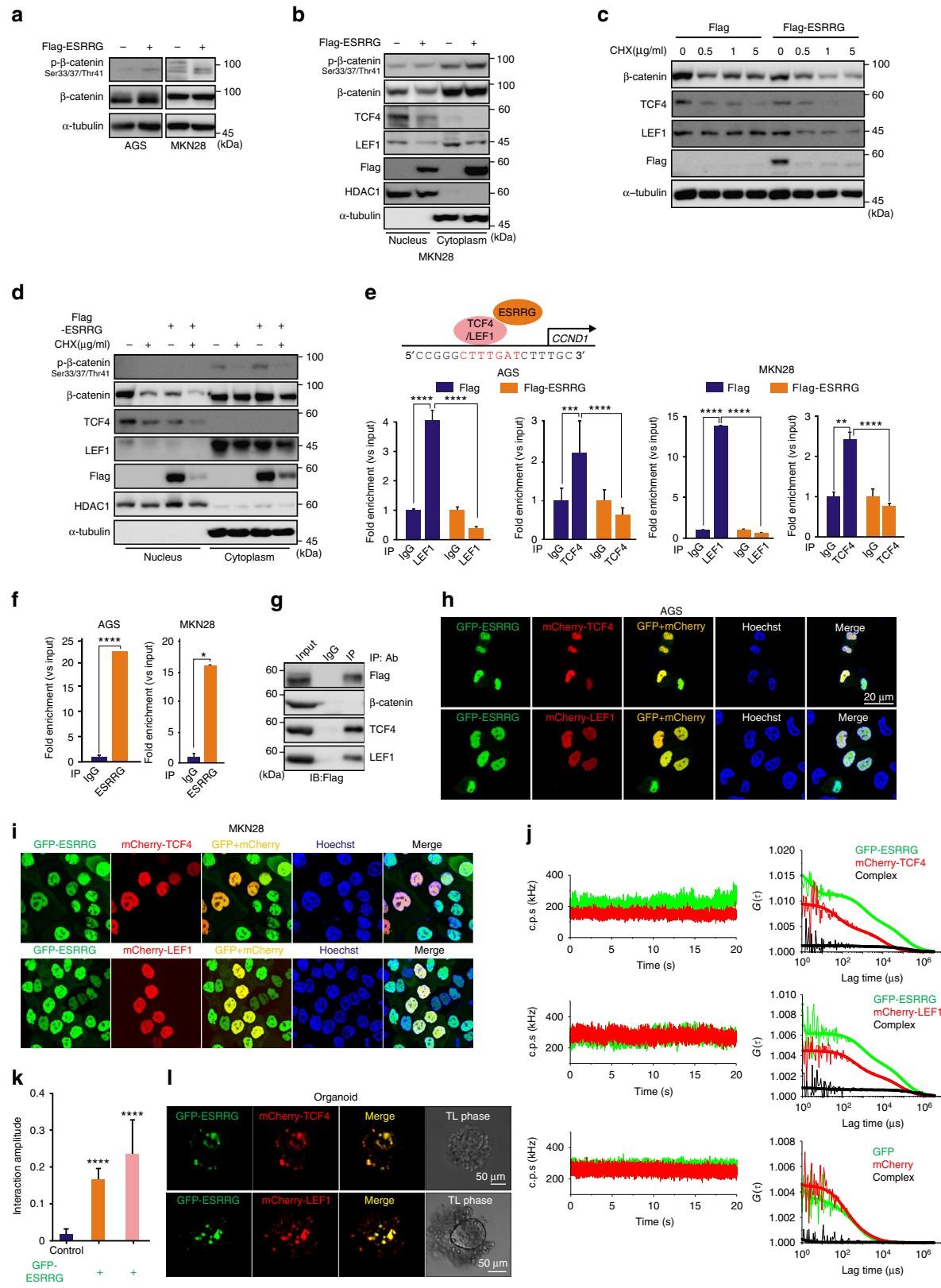

suppress Wnt signaling function and provide a potential therapeutic avenue to treat GC and potentially cancers of other lineages. The NRs governing gene transcription and expression have been recognized previously as therapeutic targets due to the nature of their physiological functions. NR agonists or antagonists are therefore widely used as cancer treatments. For example, tamoxifen, which targets ESR1 as an antagonist, is a well-known breast cancer drug[14]. Although the PPARG agonists rosiglitazone and pioglitazone are not used in the clinic, those agonists can suppress cancer cell proliferation[29]. DY131 was developed as a specific agonist of ESRRG transactivation[30]. Although DY131 also activates ESRRB, its activity appears to be mediated by ESRRG since ESRRG-deficient mice are not responsive to DY131 treatment[31,32]. DY131 treatment could prevent GC progression by enhancing ESRRG activity by suppressing Wnt signaling. Previously developed Wnt inhibitors (XAV-939, ICG-001, Wnt C59) were not effective in GC cells compared with DY131 (Supplementary Figs. 11 and 14). The development of more specific and pharmacologically tractable agonists that enhance ESRRG activity could provide novel approaches to improve GC outcomes.

NRs govern diverse signaling pathways including Wnt signaling as TFs and have overlapping functions with each other to regulate physiological functions[21]. PPAR gamma and delta have previously been shown to influence the Wnt pathway[33]. Since NRs crosstalk with each other, ESRRG shares a functional relationship with the PPARs to regulate Wnt signaling pathway. Since this relationship across the NRs has not been extensively explored in GC, further elucidation regarding crosstalk mechanisms is needed.

Tumor suppressors are frequently hypermethylated or mutated in human cancer, and oncogene activation leads to tumorigenesis[34–36]. Many TFs including *TP53*, *BRCA1*, and *RUXN3* function as tumor suppressors[31], and deregulation (methylation or mutation) of TFs leads to tumorigenesis. Recent genomic analysis from public databases suggests that ESRRG is not frequently methylated or mutated in GC (www.cbioportal.org)[36]. Thus, the decreased levels of ESRRG in GC are likely to be through a mechanism that is distinct from conventional tumor suppressive TFs. Thus, the mechanisms underlying the deregulation of ESRRG during GC progression will require further elucidation. Furthermore, while ESRRG clearly antagonizes the Wnt signaling pathway, the effects of ESRRG on other oncogenic pathways such as Notch signaling or JAG1 could contribute to the functional effects of ESRRG as well as to its therapeutic activation (Fig. 4a). How ESRRG is involved in cancer progression is thus worthy of further investigation.

In summary, we have demonstrated that ESRRG is a novel tumor suppressor that inhibits Wnt signaling in GC. We propose that ESRRG represents a novel therapeutic target for the treatment of GC.

## Methods

**Gene expression data analysis**. The gene expression data used from the NCBI GEO databases are publically available (accession numbers GSE13861[6], GSE26899, GSE29272[37], GSE62254[38]). All of these data were downloaded and processed using BRB array tools for further analysis[12].

**Microarray**. Following the overexpression of ESRRG in ASG or MKN45 cells for 3 days, the cells were harvested for RNA isolation using a mirVana™ RNA Isolation labeling kit (Ambion, Inc, Waltham, MA). The extracted total RNA (500 ng) was then used for labeling and hybridization to Human BeadChip V4 microarrays (Illumina, San Diego, CA) in accordance with the manufacturer's protocols. After the bead chips were scanned with an Illumina BeadArray Reader, the microarray data were normalized using the quantile normalization method in the Linear Models for Microarray Data package in the R language environment. The expression level of each gene was then $\log_2$ transformed before further analysis. The microarray data are available from the NCBI Gene Expression Omnibus public database (GSE78050).

**Cell lines and reagents**. GC cell lines were purchased from the American Type Culture Collection (ATCC) and Korean Cell Line Bank (KCLB). Mycoplasma test was done using MycoAlert™ Mycoplasma Detection Kit (Lonza; LT07-118). Cells were grown in Dulbecco's modified essential medium or RPMI1640 supplemented with 10% fetal bovine serum at 37 °C in a humidified incubator with 5% $CO_2$. Reagents were sourced commercially as follows: DY131 (#2266; TOCRIS, Bristol, UK), GSK5182 (#AOB1629; Aobious, Gloucester. MA), ICG-001 (#S2662), XAV-939 (#S1180), and Wnt-C59 (#S7037; Selleckchem, Houston, TX), and CHX (#01810; Sigma-Aldrich, St Louis, MO).

**Immunohistochemistry**. For immunohistochemical analysis, tissue blocks were cut into 5-μm-thick sections, deparaffinized in xylene, and rehydrated in a graded alcohol series. Antigen retrieval was performed by irradiation (microwave oven) for 20 min in a jar containing 0.01 M citrate buffer (pH 6.0) and incubation with 0.025% trypsin in 50 mM Tris buffer for 5 min. Endogenous peroxidase was blocked using 3% hydrogen peroxide in phosphate-buffered saline (PBS) for 12 min. The specimens were then incubated with a protein-blocking solution consisting of PBS (pH 7.5) with 5% normal horse serum for 30 min at room temperature. Incubation with primary antibodies was performed at 4 °C overnight. Primary antibodies against the following proteins were used at the indicated dilutions: ESRRG (1:100, PP-H6812-00; R&D systems, Minneapolis, MN), Ki67 (1:50, ab833; Abcam, Cambridge, UK), CCND1 (1:100, #2926; Cell Signaling Technology (CST), Danvers, MA), CTNNB1 (1:100, #610153; BD Biosciences, San Jose, CA), TCF4 (1:50, #2569; CST), and LEF1 (1:500, #A303-487A; Bethyl Laboratories, Montgomery, TX). The samples were then rinsed and incubated with peroxidase-conjugated anti-goat IgG for 1 h at room temperature. The slides were then rinsed with PBS and incubated for 5 min with an ImmPACT™ DAB Kit (Vector Laboratories, Burlingame, CA). The sections were next washed three times with distilled water, counterstained with Mayer's hematoxylin (Sigma-Aldrich, St Louis, MO), and washed once each with distilled water and PBS. Slides were mounted using a Universal Mount (Vector Laboratories) and examined using a brightfield microscope. ESRRG and Ki67 expression in tumor cells was assessed by independent pathologists according to previously described methods.

**Xenograft experiments**. Male or female athymic nude mice were purchased from Oriental Bio (Seoul, Korea) and maintained according to the animal experimentation guidelines of Asan Medical Center. All mouse studies were approved and supervised by the Asan Medical Center Institutional Animal Care and Use Committee (IACUC No.2015-14-178). The mice were between 8 and 12 weeks of age at the time of injection. Cells were subcutaneously injected ($4 \times 10^6$ cells in 50 μl of normal saline) to establish tumors. Treatment continued until the animal became moribund (typically 4–6 weeks following tumor cell injection). At the time of sacrifice, body weight, tumor weight, and tumor location were recorded. Tumor

**Fig. 5** ESRRG directly antagonizes Wnt signaling. **a, b** AGS or MKN28 (**b**) cells were infected with Flag or Flag-ESRRG-lentiviral vector. The cells (**a**) including fractionated samples (**b**) were also used for western blotting with the indicated antibodies. **c, d** The infected AGS (**c**) and MKN28 (**d**) cells were treated with CHX for the indicated times and the samples were used for western blotting with the indicated antibodies. **e** ChIP assays were performed on AGS or MKN28 cells after transfection with ESRRG using a TCF4/LEF1 antibody. Recruitment of ESRRG to the *CCDN1* promoter via TCF4/LEF1 was analyzed using primers specific to the *CCND1* promoter. **f** ChIP assay was done with ESRRG antibody. IgG was used as an internal control. **g** Immunoprecipitation was done in MKN28 cells with the indicated antibodies and detected with Flag antibody. **h, i, l** GC cells or organoids (**l**) were transfected with GFP-ESRRG and mCherry-TCF4 or LEF1 and used for cellular imaging under a confocal microscope. **j** Changes over time in the average fluorescence intensities (count per second; c.p.s in kHz) of GFP-ESRRG (green) and mCherry-TCF4 or LEF1 (red), and the corresponding correlation functions, are shown. Changes over time in the average fluorescence intensity and the corresponding correlation functions obtained in cells co-expressing monomer GFP and mCherry are also shown. **k** Summary of protein interaction amplitudes. The interaction amplitude represents the mean value of the relative cross-correlation amplitude. Data represent the mean ± s.d. from three independent replicates. Student's *t*-test was used to examine statistical significance (*$p < 0.05$, **$p < 0.01$, ***$p < 0.005$, ****$p < 0.001$)

tissues were snap-frozen for lysate preparation. The individuals who performed the necropsies, tumor collection, and tissue processing were blind to the treatment group assignments.

**Cell proliferation assay**. Stably or transiently transfected cells were used for cell growth assays. The proliferation assay was performed in accordance with the manufacturer's instructions (CCK8-Kit; CK04-20, Dojindo, Rockville, MD).

**Colony forming assay**. Cells were infected with lentivirus or treated with the indicated compound for a designated time. Cells (500) were seeded in 6-well plates and fixed 14 days later with 3.7% paraformaldehyde for 5 min, and stained with 0.05% crystal violet for 15 min. Colonies containing more than 50 cells were counted.

**Reporter assay**. TOP/Flash reporter, *CCND1* promoter, TCF4/TCF7L2, and LEF1 cDNAs were purchased from Addgene. ESRRG cDNA has been described

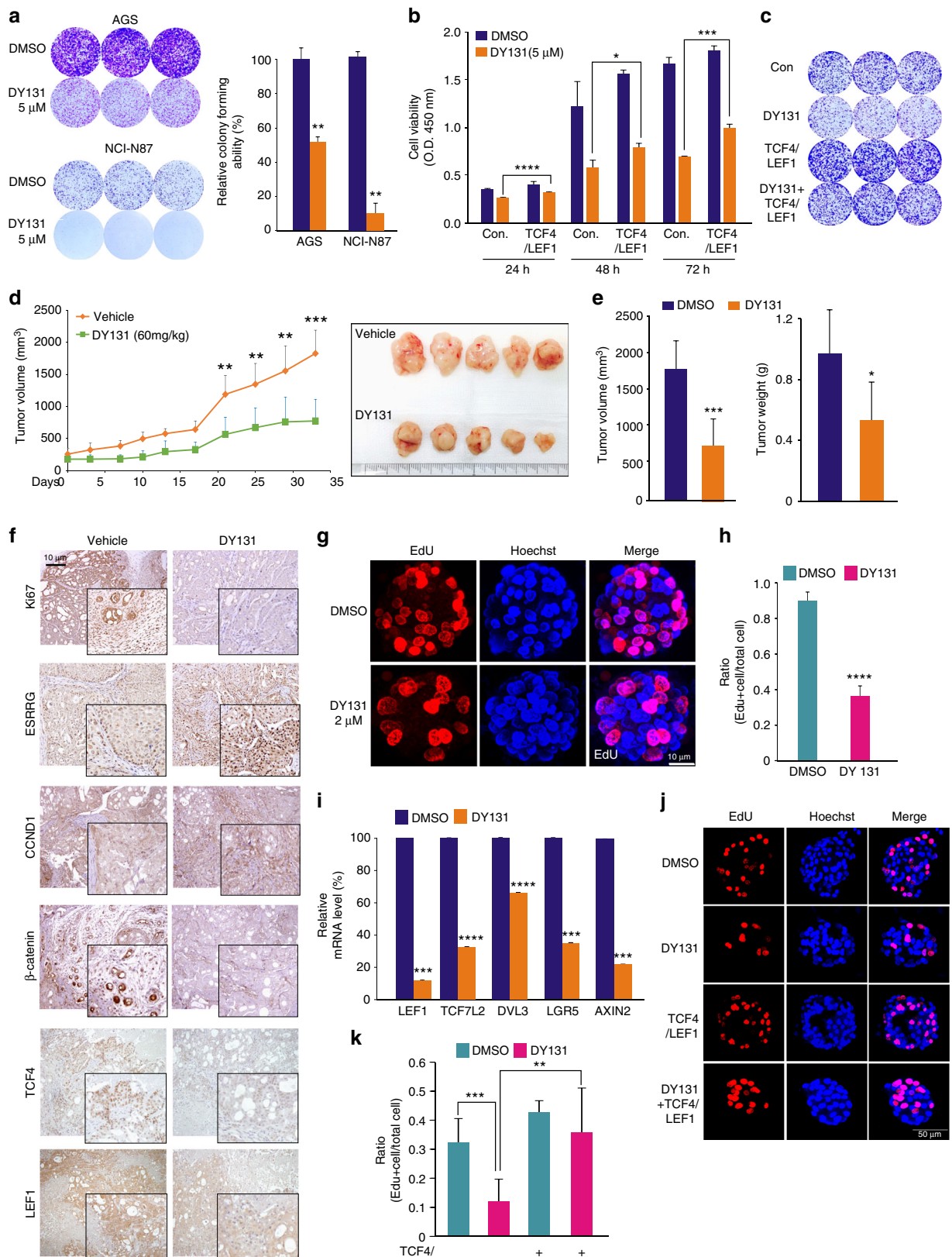

previously[39,40]. For luciferase-based reporter assays, cells were transfected with indicated reporter genes and plasmids using Lipofectamine 3000 (Invitrogen, Grand Island, NY) in accordance with the manufacturer's instructions. After 48 h, cells were harvested for measurement of luciferase activity with a Promega kit (E1605).

**ChIP assay.** ChIP assays were performed using a Thermo Scientific, Pierce[TM] Magnetic ChIP Kit (#26157) in accordance with the manufacturer's protocol with minor modifications.[7] Briefly, AGS Cells (10[7]) were used for each reaction and were treated with 1% formaldehyde for cross-linking and subsequently harvested. After sonication, 1% of the soluble chromatin fraction was de-cross-linked by heating at 65 °C overnight and used as input. The remaining chromatin fraction was immunoprecipitated with ESRRG antibody and de-cross-linked by heating. DNA was purified using the QiaQuick PCR purification kit (Qiagen, Venlo, Netherlands) and analyzed by PCR. The antibodies used were specific for ESRRG (PP-H6812-00; R&D Systems) and normal mouse immunoglobulin G (SC-2027; Santa Cruz Biotechnology, Dallas, TX). Purified DNA was used in qRT-PCR for the quantification of protein–DNA binding with a SensiFAST SYBR Hi-ROX kit (#BIO-92005; Bioline). The primer sequences were as follows: forward 5′ GGGCGATTTGCATTTCTATG 3′, reverse 5′ACTCCCCTGTAGTCCGTGTG3′.

**Immunoprecipitation.** Cell lysates were prepared using NP-40 lysis buffer (10% glycerol, 0.5% nonidet P-40, 125 mM NaCl, 1 mM EDTA, 20 mM Tris-Cl (pH 8.0)). Lysates were precipitated with Dynabeads Protein G (Thermo, Rockford, IL) overnight at 4 °C. Precipitates were washed 3 times with lysis buffer and then boiled in 2× SDS-PAGE sample buffer prior to immunoblotting.

**Preparation of nuclear and cytoplasmic fractions.** Nuclear and cytoplasmic extractions were performed using an NE-PER Nuclear Cytoplasmic Extraction Reagent kit (Thermo, Rockford, IL) according to the manufacturer's instructions. Briefly, the treated cells were washed twice with cold PBS and centrifuged at 500×g for 3 min. The cell pellet was then suspended in 200 µl of cytoplasmic extraction reagent I by vortexing. The suspension was subsequently incubated on ice for 10 min followed by the addition of 11 µl of a second cytoplasmic extraction reagent II, vortexing for 5 s, incubation on ice for 1 min, and centrifugation for 5 min at 16,000×g. The supernatant (cytoplasmic fraction) was transferred to a pre-chilled tube. The insoluble pellet fraction, which contains crude nuclei, was resuspended in 100 µl of nuclear extraction reagent by vortexing for 15 s every 10 min over a total period of 40 min, and then centrifuged for 10 min at 16,000×g. The resulting supernatant constituted the nuclear extract.

**Western blotting.** Western blot analysis was performed as described previously using antibodies against ESRRG (#H6812; R&D Systems), total and phosphor-β-catenin (#9561 and #9562; Cell Signaling Technology (CST), Danvers, MA), TCF4 (#2569;CST), Flag (#2368;CST), LEF1 (#A303-487A; Bethyl Laboratories), HDAC1 (#7872; Santa Cruz Biotechnology), β-actin (#A5316; Sigma-Aldrich), pGSK3α/β (#9331; CST), GSK3α/β (#ab15314; Abcam), and α-tubulin (#3873; CST). Antibodies were diluted with bovine serum albumin (BSA) and used at a 1:1000 ratio on membranes blocked with BSA. Please see Supplementary Fig. 16 for uncropped scans of western blots.

**qRT-PCR.** Total RNA was extracted from the indicated cell lines or patient samples using a mirVana RNA isolation kit (Ambion) in accordance with the manufacturer's instructions and analyzed by real-time qRT-PCR with TaqMan primers specific for each gene of interest (ABI). Real-time PCR was performed using the StepOne[TM] Real-Time PCR system with a 96-well block module (ABI). Cycling conditions were 45 °C for 30 min and 95 °C for 10 min, followed by 40 cycles of 95 °C for 15 s and 60 °C for 60 s. The relative amounts of mRNA were calculated from the threshold cycle number using the cyclophilin A (PPIA) expression as a housekeeping control. All experiments were performed in triplicate and the values obtained were averaged.

**Lentiviral transduction.** An ESRRG lentiviral expression vector was constructed by cloning its full-length cDNA fragment into pCDH-EF1-T2A-Puro using an infusion system (SBI, Mountain View, CA). To produce lentiviral particles, this vector was co-transfected with the lentiviral packaging plasmids pLP1, pLP2, and pLP/VSVG (Invitrogen) into 293FT cells. Lipofectamine 2000 (Invitrogen) was

used as the transfection reagent. At 48–72 h post transfection, the virus-containing cell culture media was harvested and frozen in aliquots. A moderate multiplicity of infection (MOI = 3) was used for the transduction of GC cells to minimize negative effects on cellular proliferation. All experiments were performed at 2–4 days after infection,.

**Organoid culture.** Organoid cultures were generated based on a previous report[41] with minor modifications. Briefly, gastric fundus organoids were derived from surgical samples and informed consent was received from GC patients at Yonsei University Severance Hospital (IRB No. 4-2015-0877). The sampled gastric tissue was then mixed in matrigel (BD Biosciences, San Jose, CA) in culture. Culture conditions included Advanced Dulbecco's modified Eagle medium/F12 medium (Invitrogen, Carlsbad, CA), Wnt-conditioned medium, R-spondin-conditioned medium supplemented with gastric growth factors including bone morphogenetic protein inhibitor, noggin (PeproTech, Rocky Hill, NJ), GlutaMAX-I (Invitrogen, Carlsbad, CA), B27 (Invitrogen, Carlsbad, CA), TGF beta I A83-01 (TOCRIS, Bristol, UK), Nicotinamide, N-acetylcysteine, ROCK I Y-27632, gastrin (Sigma-Aldrich, St Louis, MO), epidermal growth factor (PeproTech), and fibroblast growth factor 10 (R&D systems, Minneapolis, MN). The cells matured into organoids after 1–2 days. Gastric organoids were subsequently passaged every 12 days.

**5-ethynyl-2′-deoxyuridine labeling.** To identify and count the proliferating cells, GC organoids were analyzed using a Click-iT EdU imaging Kit (C10340; Invitrogen, Carlsbad, CA) in accordance with the manufacturer's instructions. Briefly, organoids were incubated in 5 µM EdU in Opti-MEM for 1 h at 37 °C. The Click iT reaction cocktail was then added to the cells as described in the protocol and incubated for 30 min at room temperature, followed by two washes in PBS. Organoids were then incubated with the Hoechst 33342 DNA dye at a dilution of 1:2000 in Opti-MEM for 30 min.

**Confocal laser scanning microscopy and live cell imaging.** Fluorescence observations were undertaken using an LSM780 inverted confocal laser scanning microscope (LSM: Carl Zeiss, Jena, Germany) at room temperature. The z-stack profiles (total stack size, 80 µm) were acquired at 2.00-µm intervals from the bottom to the top of the organoid. Microscopy images were processed and analyzed using ZEN2012 software installed on the LSM780 microscope.

**Fluorescence cross-correlation spectroscopy.** Dual-color FCCS measurements were all performed at 25 °C with an LSM780 confocal microscope (Carl Zeiss, Germany) as described previously[19,42]. Briefly, FCCS setups using the LSM780 microscope consisted of a continuous-wave Ar+ laser (25 mW) and a solid-state laser (20 mW), a water-immersion objective (C-Apochromat, ×40/1.2 NA; Carl Zeiss), and two channels of a GaAsP multichannel spectral detector (Quasar; Carl Zeiss). GFP was excited with the 488-nm laser line and mCherry with the 561-nm laser line, with a minimal total power to allow an optimal signal-to-noise ratio. The confocal pinhole diameter was adjusted to 37 µm for the 488- and 561-nm lasers. Emission signals were split by a dichroic mirror (488/561-nm beam splitter) and detected at 500–550 nm in the green channel for GFP and at 600–690 nm in the red channel for mCherry. FCCS data were analyzed using the analytical component of the ZEN 2012 acquisition software (Carl Zeiss). Briefly, the fluorescence auto-correlation functions of the red and green channels, $G_r(\tau)$ and $G_g(\tau)$, and the fluorescence cross-correlation function, $G_c(\tau)$, were calculated from

$$G_x(\tau) = 1 + \frac{\langle \delta I_i(t) \cdot \delta I_j(t+\tau) \rangle}{\langle I_i(t) \rangle \langle I_j(t) \rangle}, \tag{1}$$

where $\tau$ denotes the time delay, $I_i$ the fluorescence intensity of the red channel ($i =$ r) or green channel ($i =$ g), and $G_r(\tau)$, $G_g(\tau)$, and $G_c(\tau)$ denote the fluorescence auto-correlation functions (FAFs) of red ($i = j = x =$ r), green ($i = j = x =$ g), and fluorescence cross ($i =$ r, j $=$ g, x $=$ r) correlation function (FCF), respectively. The acquired $G_x(\tau)$ values were fitted using a one-, two-, or three-component model:

$$G_x(\tau) = 1 + \frac{1}{N} \sum_i F_i \left(1 + \frac{\tau}{\tau_i}\right)^{-1} \left(1 + \frac{\tau}{s^2 \tau_i}\right)^{-1/2}, \tag{2}$$

where $F_i$ and $\tau_i$ are the fraction and diffusion time of component $i$ respectively. $N$ is the average number of fluorescent particles in the excitation–detection volume

**Fig. 6** Therapeutic efficacy of the DY131 ESRRG agonist in GC. **a** The indicated cells were treated with DY131 at the dose shown and colony formation assays were performed. AGS (**b**) and MKN28 (**c**) cells were treated with DY131 and then transfected with TCF4/LEF1 and used in a cell proliferation (**b**) and colony formation (**c**) assay. **d** After NCI-N87 cell implantation, DY131 or vehicle was intraperitoneally injected into mice every 3 days and the tumor volume was measured at the indicated time points (n = 5 per group). **e** Tumor volumes and weights were measured from sacrificed mice. **f** Immunohistochemical analysis of mouse samples. **g–k** Organoids from GC patients were incubated with DY131 and TCF4/LEF1 or without for 72 h and then stained with EdU and Hoechst dye. **g**, **k** Proliferation from organoids was quantified using Edu staining. **i** qRT-PCR analysis of human organoid samples. Data represent the mean ± s.d. from three independent replicates. Student's t-test was used to examine statistical significance (*p < 0.05, **p < 0.01, ***p < 0.005, ****p < 0.001)

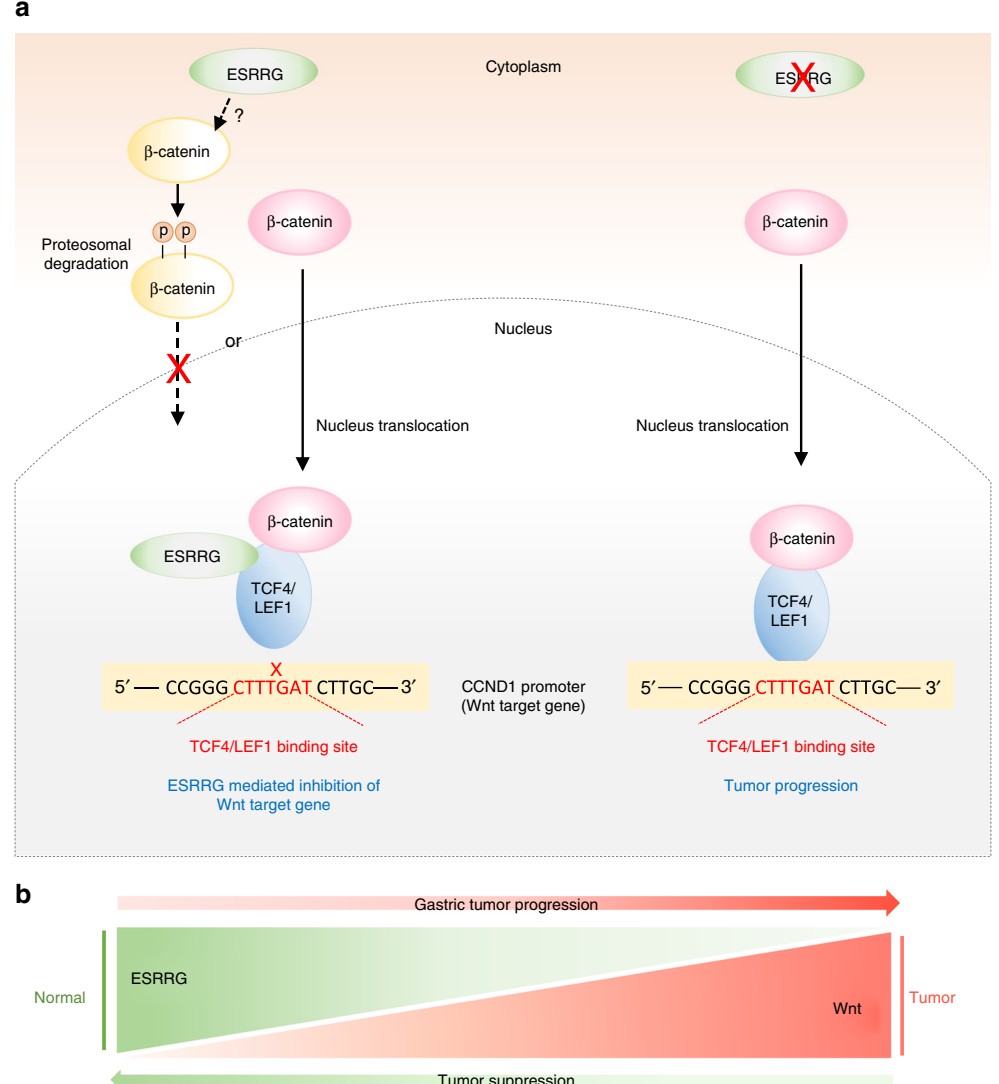

**Fig. 7** Schematic diagram of the gene regulation of ESRRG and Wnt signaling. **a** ESRRG induces β-catenin degradation via an unknown mechanism and inhibits the binding of the Wnt component TCF/LEF to the *CCND1* gene promoter region by direct interaction, thus potentially repressing the Wnt signaling pathway and blocking tumorigenesis. **b** The balance of ESRRG and Wnt signaling activity is crucial for tumor suppression or formation in GC

defined by the radius $w_0$ and the length $2z_0$, and $s$ is the structure parameter representing the ratio $s = z_0/w_0$. The structure parameter was calibrated using Rhodamine-6G (Rh6G) solution. The positions for FCCS measurements were selected in the nuclei of cells. All measured FAFs from live cells were globally fitted with the software installed on the LSM780 system using the two-component model ($i = 2$) with or without a triplet term to estimate the diffusion coefficient. For simplicity, the triplet term in Eq. (2) was not shown. For the evaluation of the interaction amplitude, the amplitude of the cross-correlation function was normalized to the amplitude of the autocorrelation function of GFP or mCherry to calculate the relative cross-correlation amplitude (RCA; $[G_c(0)-1]/[G_r(0)-1]$) corresponding to the fraction of associated molecules ($N_c/N_g$). As a negative control, FCCS measurements were performed using cells co-expressing GFP and mCherry.

**Wnt/β-catenin signaling assays**. AGS or MKN28 cells were seeded into 6-well plates and cultured until reaching 60–70% confluence. The cells were then transiently transfected with either Flag or Flag-ESRRG and the β-catenin MT (S37A). Total β-catenin and phosphorylated β-catenin were quantified using ELISA (#85-96143-11, Instant One ELISA, affymetrix eBioscience, Grand Island, NY) in accordance with the manufacturer's instructions.

**Statistical analysis and survival analysis**. The random variance *t*-test was applied to identify genes differentially expressed between the two classes using Biometric Research Branch (BRB) ArrayTools (National Cancer Institute, Bethesda, MD). Gene expression differences were considered statistically significant

if the *p*-value was less than 0.001. Cluster analysis was performed with Cluster and Treeview. Kaplan–Meier plots and log-rank test were used to estimate patient prognoses.

**Data availability**. The genomic data are available from the NCBI Gene Expression Omnibus (GEO) under accession numbers (GSE13861[6], GSE26899, GSE29272[37], GSE62254[38], and GSE78050). Extra data are available from the corresponding author on request

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

## Acknowledgements

This research was supported by the Ministry of Education (NRF-2017R1A2B4001962 to Y.-Y.P.) and by a grant of the Korea Health Technology R&D Project through the Korea Health Industry Development Institute (KHIDI), funded by the Ministry of Health & Welfare, Republic of Korea (HI15C0972 and HI17C0380 to Y.-Y.P.). We thank the Confocal Microscope core facility at the ConveRgence mEDIcine research cenTer (CREDIT), Asan Institute for Life Science, for support and instrumentation.

## Author contributions

M.-H.K. and Y.-Y.P. generated the hypothesis, designed the experiments, and wrote the manuscript. M.-H.K. performed the experiments. M.-H.K., H.C., M.O., J.-H.J., S.K., J.H. L., Y.S.P., H.-S.C., M.-N.K., C.-G.P., J.-S.L., G.B.M., S.-J.M., and Y.-Y.P. interpreted the data. M.-H.K., G.B.M., and Y.-Y.P. edited the manuscrip. All authors approved the final draft.

## Additional information

**Competing interests:** The authors declare no competing interests.

