## [Peer Review File · Nature Communications]

Reviewers' comments:

Reviewer #1 (Remarks to the Author):

The manuscript by Kang M-H. et al. describes ESRRG (ERR γ) as a novel tumor suppressive gene in gastric cancer. Using publically available gene expression data, the authors found that ESRRG is down-regulated in gastric tumor tissues as compared with non-tumor tissues. Overexpression of ESRRG suppressed proliferation of gastric cancer cells in vitro and in vivo, and patients with a higher expression of ESRRG had good clinical outcomes across multiple sample sets from GEO. The authors also found that genes whose expression was down-regulated in ESRRG-overexpressing AGS cells include oncogenic genes, such as CCND1 and PCNA, and Wnt signal-related genes, such as DVL3, LEF1, and TCF7L2. The inhibition of gastric cancer cell proliferation by ESRRG overexpression was compensated by the expression of TCF7L2 and LEF1. Furthermore, the authors showed that ESRRG increases phosphorylated β -catenin level and blocks LEF1-binding to the CCND1 promoter. Finally they demonstrated that an ESRRG agonist DY131 inhibits gastric cancer cell proliferation in vitro and in vivo. From these results, the authors concluded that ESRRG functions as a negative regulator of Wnt signaling in gastric cancer and ESRRG may be a therapeutic candidate for gastric cancer therapy.

The authors' findings that ESRRG inhibits the Wnt-signaling pathway and down-regulation of ESRRG is related with poor prognosis of gastric cancer are interesting. The weakness in this study is the lack of mechanistic analysis how ESRRG expression inhibits Wnt signaling. The authors need to address several concerns to support their conclusions. The comments are as follows:

Comments

1. In Fig. 2a, b and Supplementary Fig. 4 the authors showed that ESRRG overexpression inhibits in vitro proliferation of AGS, NCI-N87, MKN28, and MKN45 cells. In Fig. 2c-f, one more cell line in addition to NCI-N87 cell should be tested for the inhibition of xenograft tumor formation by ESRRG overexpression.
2. The differences between control and ESRRG-expressing cells are very small in Fig. 4c and the data is not convincing. The authors should test effects of TCF7L2 and LEF1 on the ESRRG-induced inhibition of gastric cancer cell proliferation using colony formation and xenograft tumor formation assays to clearly show that Wnt signaling downregulation is really involved in ESRRG-dependent cell proliferation inhibition. These experiments are essential for supporting the authors' conclusion.
3. The mechanism by which ESRRG inhibits Wnt signaling is unclear. Since ESRRG increased the phosphorylation state of β -catenin, does ESRRG de-stabilize β -catenin? If so, the authors should examine the amounts of cytoplasmic β -catenin and intranuclear β -catenin biochemically in Fig. 4e. This reviewer does not understand the aim of experiments and the interpretation of results in Fig. 4f.
4. Fig. 4g suggested that ESRRG competes with Myc-LEF for binding to the CCND1 promoter. Does ESRRG interact with LEF or the CCND1 promoter directly? The detailed mechanism should be clarified. Taken together with Fig. 4e and f, the authors must show whether ESRRG affects the β -catenin or LEF1/TCF level to inhibit Wnt signaling.
5. The results in Fig. 5 are interesting but must be interpreted carefully, because DY131 activates transcriptional activity of ESRRG and ERR β . The data indeed showed that DY131 inhibits gastric cancer cell proliferation, but whether the inhibition is due to the suppression of Wnt signaling is not clear. In Fig. 5d CTNNB1 was decreased by DY131 in the tumor lesion. The reasons should be explained. The author described that CTNNB1 is a Wnt downstream target gene in the text (lines 209 and 210), what

does this mean? β -catenin is an important component of Wnt signaling but not a downstream target of Wnt signaling. Although Wnt-related genes are downregulated by DY131 in Fig. 5f, the results do not always support that the cell proliferation suppression by DY131 is through the inhibition of Wnt signaling.

Reviewer #2 (Remarks to the Author):

The authors found that the transcription factor ESRRG, or estrogen-related receptor gamma, behaves as a tumor suppressor in gastric cancer. It does so by inhibiting Wnt signaling. The authors find that ESRRG is downregulated in tumor vs normal gastric tissue and that activation of ESRRG inhibits tumor growth.

This is a well-written manuscript with some interesting findings linking ESRRG to Wnt signaling. The conclusions however, cannot be supported for the following reasons:

The rationale for focusing on ESRRG is not clear, given that the N/T differential signature showed many other transcription factors that were also differentially expressed.

In TCGA no deletion of ESRRG are found, only mutations in ~5% and even rare amplifications (2%). This contradicts what's stated in the discussion and does not support a role for this transcription factor as a tumor suppressor.

In terms of gene expression profiling, while ESRRG loss of expression was consistently associated with poorer prognosis, this may be due to other changes in the transcriptome that co-occur with ESRRG loss.

While the proposed mechanism of action is centered on Wnt signaling ESRRG has also been shown to modulate proliferation via DNMT1. This mechanism is not explored. Wnt signaling in gastric cancer has been explored extensively and inhibition of this pathway has not been successful. It is therefore unlikely that activation of an upstream regulator of this pathway (and of other ones) will be successful. At a minimum, a comparison between the ESRRG activator and DKN-01 should be attempted. The experiment in figure 4 C, the rescue of Wnt downstream effectors on ESRRG overexpression is minimal and of a smaller magnitude than the difference in control cells (blue bars) in the FLAG vs FLAG- ESRRG.

Overlapping function with other nuclear hormone receptors is well known as well and signaling in these pathways is not addressed. Furthermore, all experiments were carried out in male nude mice. No female mice were used.

Reviewer #3 (Remarks to the Author):

Kang et al. describe a tumor suppressive effect of the nuclear receptor ERRgamma in gastric cancer. The data clearly shows decreased expression in tumor relative to non-tumor samples, and decreased expression is correlated with poorer clinical outcomes. Conversely ERRgamma overexpression or agonist activation decreases growth of gastric cancer cell lines and tumor xenografts. Anti-proliferative effects of ERRgamma are linked to inhibition of beta-catenin activity.

The primary concern is that the basis for the antagonism of beta-catenin by ERRgamma is quite unclear. It is suggested that beta-catenin phosphorylation is altered, but the data supporting this is

not compelling. It is also suggested that binding of overexpressed LEF to a site in the cyclin D1 promoter is decreased by ERRgamma overexpression, but the basis for this is unclear, and this experiment should have been done with endogenous LEF. ERRgamma overexpression also strongly decreases transactivation of the Top Flash reporter, but the basis for this effect is not clear. Overall, the impact of ERRgamma on Wnt signaling is clear, but the lack of a clear mechanism to account for this leaves the story incomplete.

Reviewers' comments:

Reviewer #1 (Remarks to the Author):

The manuscript by Kang M-H. et al. describes ESRRG (ERRγ) as a novel tumor suppressive gene in gastric cancer. Using publically available gene expression data, the authors found that ESRRG is down-regulated in gastric tumor tissues as compared with non-tumor tissues. Overexpression of ESRRG suppressed proliferation of gastric cancer cells in vitro and in vivo, and patients with a higher expression of ESRRG had good clinical outcomes across multiple sample sets from GEO. The authors also found that genes whose expression was down-regulated in ESRRG-overexpressing AGS cells include oncogenic genes, such as CCND1 and PCNA, and Wnt signal-related genes, such as DVL3, LEF1, and TCF7L2. The inhibition of gastric cancer cell proliferation by ESRRG overexpression was compensated by the expression of TCF7L2 and LEF1. Furthermore, the authors showed that ESRRG increases phosphorylated β-catenin level and blocks LEF1-binding to the CCND1 promoter. Finally they demonstrated that an ESRRG agonist DY131 inhibits gastric cancer cell proliferation in vitro and in vivo. From these results, the authors concluded that ESRRG functions as a negative regulator of Wnt signaling in gastric cancer and ESRRG may be a therapeutic candidate for gastric cancer therapy.

The authors' findings that ESRRG inhibits the Wnt-signaling pathway and down-regulation of ESRRG is related with poor prognosis of gastric cancer are interesting. The weakness in this study is the lack of mechanistic analysis how ESRRG expression inhibits Wnt signaling. The authors need to address several concerns to support their conclusions. The comments are as follows:

Comments

1. In Fig. 2a, b and Supplementary Fig. 4 the authors showed that ESRRG overexpression inhibits *in vitro* proliferation of AGS, NCI-N87, MKN28, and MKN45 cells. In Fig. 2c-f, one more cell line in addition to NCI-N87 cell should be tested for the inhibition of xenograft tumor formation by ESRRG overexpression.

Response: As the reviewer requested, we performed xenograft experiments using another GC cell line, MKN45 (Fig. 4e). As shown in Figure 4e, MKN45 cells, with or without ectopic overexpression of ESRRG, were subcutaneously transplanted into athymic nude mice, and tumor growth was monitored. As expected, ESRRG suppressed tumor growth in this *in vivo* mouse model (Fig. 4e-f) consistent with Figure 2c. Our *in vivo* experiments thus collectively demonstrate that ESRRG plays a generalized tumor suppressive role in GC.

2. The differences between control and ESRRG-expressing cells are very small in Fig. 4c and the data is not convincing. The authors should test effects of TCFL2 and LEF1 on the ESRRG-induced inhibition of gastric cancer cell proliferation using colony formation and xenograft tumor formation assays to clearly show that Wnt signaling down-regulation is really involved in ESRRG-dependent cell proliferation inhibition. These experiments are essential for supporting the authors' conclusion.

Response: As requested we have replaced Figure 4c with new data. Our new cell proliferation and colony formation assays showed that growth inhibition by ESRRG was rescued by the re-introduction of TCF4/LEF1 (Fig. 4c and d). In addition, we performed a xenograft experiment using MKN45 cells as the reviewer suggested. As now shown in Figure 4e and f, *in vivo* tumor growth inhibited by ESRRG was rescued by the Wnt components, TCF4 and LEF1. Collectively, our data now clearly demonstrate that ESRRG suppresses cell growth by inhibiting the Wnt signaling pathway *in vitro* and *in vivo*.

3. The mechanism by which ESRRG inhibits Wnt signaling is unclear. Since ESRRG increased the phosphorylation state of β -catenin, does ESRRG de-stabilize β -catenin? If so, the authors should examine the amounts of cytoplasmic β -catenin and intranuclear β -catenin biochemically in Fig. 4e. This reviewer does not understand the aim of experiments and the interpretation of results in Fig. 4f.

Response: The data in original Figures 4e and 4f are now shown in Figure 5a and Supplementary Figure 10a. In the revised manuscript, we have performed additional experiments to provide a more detailed mechanism of how ESRRG inhibits Wnt signaling. As the reviewer suggested, we examined the stabilization of Wnt components including β -catenin, TCF4 and LEF1. As shown in Figure 5b, the expression of these factors was decreased in the nuclear fraction, but was relatively unchanged in the cytoplasmic fraction, by expression of ESRRG. In addition, in cells treated with cycloheximide, β -catenin, TCF4 and LEF1 proteins were more rapidly destabilized by ESRRG overexpression. In addition, we identified that the destabilization of the Wnt components was due to transcriptional inhibition by ESRRG via its direct interaction with TCF4/LEF1 (Fig. 5g-k). Thus ESRRG appears to inhibit Wnt signaling through a concerted set of mechanisms including transcriptional and post translational processes.

Therefore, we provide mechanistic insights i.e. that ESRRG directly inhibits TCF4/LEF1 via a direct association. Since β -catenin forms a complex with TCF4/LEF1, its activity might be indirectly influenced by ESRRG (please see our response below in this regard). We also clearly demonstrate from our current experiments that by physically interacting with TCF4/LEF1, ESRRG inhibits the binding of TCF4/LEF1 to the *CCND1* promoter (Fig. 5e-f) and ESRRG inhibits transcriptional activity of TCF/LEF1 (Fig. 4g and h). As the reviewer suggests, we

performed ELISA to detect the phosphorylation of β -catenin since it is a more precise and sensitive method for validating this than western blotting.

We thus conclude that ESRRG functions as a tumor suppressor by antagonizing Wnt signaling via the suppression of TCF4/LEF1 gene transcription and expression.

4. Fig. 4g suggested that ESRRG competes with Myc-LEF for binding to the *CCND1* promoter. Does ESRRG interact with LEF or the *CCND1* promoter directly? The detailed mechanism should be clarified. Taken together with Fig. 4e and f, the authors must show whether ESRRG affects the β -catenin or LEF1/TCF level to inhibit Wnt signaling.

Response: As the reviewer alludes to, we performed these experiments to provide some mechanistic insights into the direct relationship between TCF4/LEF1 and the *CCND1* promoter (note: Fig. 4 has now been replaced with Fig. 5). We first did immunoprecipitation analysis to detect the direct interaction between ESRRG and TCF4/LEF1 in GC cells. In addition, to confirm this direct interaction with more precision, we used dual-color fluorescence cross-correlation spectroscopy (FCCS), which is an advanced technique for directly detecting protein interactions in live cells based on confocal microscopy (Fig. 5j and k). Our IP data in Figure 5g show that ESRRG directly interacts with TCF4 and LEF1, but not β -catenin, and our FCCS data confirm that ESRRG directly interacts with TCF4 and LEF1 in GC cells (Fig. 5j and k). In addition, confocal microscope experiment with patient organoids reveals that ESRRG was co-localized with TCF4/LEF1 (Fig. 5k). Figures 5h, i, j, k and l show that the interaction between ESRRG and TCF/LEF is significant. Next, we used ChIP assays to detect an endogenous ESRRG interaction with TCF4 and LEF1 on the *CCND1* promoter using TCF4/LEF1 antibodies. Figure 5e clearly indicates that endogenous TCF4/LEF1 bound to the *CCND1* promoter and that this

interaction was inhibited by ESRRG. When we performed ChIP analysis using ESRRG antibodies, the *CCND1* promoter region bound to TCF4/LEF1 was recruited. This confirmed that ESRRG directly interacts with a TCF4/LEF1-bound *CCND1* promoter (Fig. 5f). We could not however map the direct binding site for ESRRG on the *CCND1* promoter region within -3000bp. A reporter assay revealed that ESRRG did not activate the *CCND1* promoter (Fig. 4h) suggesting that it does not directly bind this promoter. In accordance with the reviewer's request, we now show the β -catenin, TCF4 and LEF1 levels in cells expressing ESRRG (Fig. 5b-d). Our data clearly suggest that ESRRG directly antagonizes the Wnt components TCF4 and LEF1 in GC cells.

5. The results in Fig. 5 are interesting but must be interpreted carefully, because DY131 activates transcriptional activity of ESRRG and $ERR\beta$. The data indeed showed that DY131 inhibits gastric cancer cell proliferation, but whether the inhibition is due to the suppression of Wnt signaling is not clear. In Fig. 5d CTNNB1 was decreased by DY131 in the tumor lesion. The reasons should be explained. The author described that CTNNB1 is a Wnt downstream target gene in the text (lines 209 and 210), what does this mean? β -catenin is an important component of Wnt signaling but not a downstream target of Wnt signaling. Although Wnt-related genes are downregulated by DY131 in Fig. 5f, the results do not always support that the cell proliferation suppression by DY131 is through the inhibition of Wnt signaling.

Response: The reviewer legitimately queries whether cell growth inhibition by DY131 is actually due to Wnt-signaling suppression. To test this, we performed rescue experiments that are now shown in Figures 6b, 6c, 6j, 6k and Supplementary Fig. 12b. These CCK8 and colony formation assay data indicate that the anti-proliferative effects of DY131 can be rescued by the re-

introduction of TCF4/LEF1 in GC cells and organoids-indicating that Wnt signaling inhibition by DY131 is due to suppressing TCF4/LEF1. CTNNB1 was found to be decreased by DY131 in the tumor lesion. DY131 also activated ESRRG transcription and expression (Supplementary Fig.13). ESRRG was shown to be expressed in the tumors and to suppress CTNNB1. As the reviewer has correctly pointed out, β -catenin is not a downstream target of Wnt but a Wnt component. We have corrected the relevant descriptions of this in the manuscript.

Reviewer #2 (Remarks to the Author):

The authors found that the transcription factor ESRRG, or estrogen-related receptor gamma, behaves as a tumor suppressor in gastric cancer. It does so by inhibiting Wnt signaling. The authors find that ESRRG is downregulated in tumor vs normal gastric tissue and that activation of ESRRG inhibits tumor growth.

This is a well-written manuscript with some interesting findings linking ESRRG to Wnt signaling. The conclusions however, cannot be supported for the following reasons:

The rationale for focusing on ESRRG is not clear, given that the N/T differential signature showed many other transcription factors that were also differentially expressed.

Response: Based on our genomic analysis, we selected ESRRG as a candidate tumor suppressor in GC as it showed the lowest expression among the many transcription factors we screened in the tumor tissue compared with normal tissue (normal vs tumor tissues; Fold ratio -14.851 in GSE29272; -16.514 Fold in GSE26899; -23.608 Fold in GSE13861). In addition, since ESRRG has a ligand binding pocket, it was a valid potential drug target. We have added a

sentence to the revised manuscript to more clearly outline these reasons, i.e. why we focused on ESRRG (Page 4, Line 9).

In TCGA no deletion of ESRRG are found, only mutations in ~5% and even rare amplifications (2%). This contradicts what's stated in the discussion and does not support a role for this transcription factor as a tumor suppressor.

Response: As the reviewer mentions, ESRRG is not mutated and not frequently amplified in GC. Transcription factors functioning as tumor suppressors are frequently methylated or mutated which can lead to tumorigenesis. However, ESRRG is neither mutated nor methylated. Thus, the tumor suppressive mechanism of ESRRG loss is distinct from the conventional pathways that transcription factors functioning as tumor suppressors typically utilize. We have revised this sentence in the manuscript to avoid any further misunderstanding (page 13, Line 1).

In terms of gene expression profiling, while ESRRG loss of expression was consistently associated with poorer prognosis, this may be due to other changes in the transcriptome that co-occur with ESRRG loss.

Response: Depending on the ESRRG expression level, the clinical outcomes in GC differ. Since ESRRG functions as a transcription factor, it dictates the expression of many downstream target genes. We identified that ESRRG suppresses genes involved in cell proliferation including Wnt signaling-associated genes. Patients with high ESRRG expression therefore show better clinical outcomes since a potentially oncogenic transcriptome, including Wnt signaling factors, is downregulated. Transcriptome profiling has shown that cell proliferation-related genes (CCND1, PCNA, TOP2B, SKP1 JAG1. etc) are also inhibited by ESRRG. There is a possibility therefore that diverse signaling pathways might be involved in the tumor suppressor activity and prognostic utility of ESRRG. Although ESRRG governs many signaling pathways, we focused

on Wnt signaling since ESRRG strongly influences a subset of the genes involved in the Wnt pathway.

While the proposed mechanism of action is centered on Wnt signaling ESRRG has also been shown to modulate proliferation via DNMT1. This mechanism is not explored. Wnt signaling in gastric cancer has been explored extensively and inhibition of this pathway has not been successful. It is therefore unlikely that activation of an upstream regulator of this pathway (and of other ones) will be successful. At a minimum, a comparison between the ESRRG activator and DKN-01 should be attempted. The experiment in figure 4 C, the rescue of Wnt downstream effectors on ESRRG overexpression is minimal and of a smaller magnitude than the difference in control cells (blue bars) in the FLAG vs FLAG- ESRRG.

Response: As the reviewer mentions, a previous report has suggested that SHP/NR0B2 suppresses DNMT1 expression via ESRRG (Ref. Zhang Y and Wang L., FEBS Letters 585 (2011) 1269-1275). However, while those authors did show that DNMT1 was suppressed by NR0B2 via ESRRG based on molecular analyses, they did not demonstrate cell growth suppression by ESRRG or NR0B2 and it was unclear from their findings whether ESRRG directly influenced DNMT1 signaling to dictate cancer cell proliferation. Our current microarray data in GC revealed that DNMT1 expression was not affected by ESRRG. Thus, we did not address any possible interaction between DNMT1 signaling and ESRRG in cancer cells and we focused instead on elucidating ESRRG functions in suppressing Wnt signaling.

The reviewer mentions that inhibiting Wnt signaling has not been successful previously in GC and asked us to evaluate the efficacy of ESRRG activator (DY131) and DKK1 (Wnt-negative regulator) in GC cells. DKK1 is indeed well known as a Wnt signaling inhibitor and DKN01 antibodies are currently in clinical trials. We attempted to source the DKN01 antibody commercially but it is unfortunately not available. We instead compared commercially available

Wnt inhibitors (XAV-939, ICG-001, Wnt C59) with DY131. Interestingly, while DY131 was effective in GC cells, Wnt inhibitors had limited effects in GC cells (Supplementary Fig. 14). This indicated that the ESRRG activator DY131 has good efficacy in GC cells.

Our rescue experiments were re-performed and the results revealed that growth inhibition by ESRRG was significantly rescued by TCF4/LEF1 (Fig. 4 c-f). Thus targeting ESRRG has the potential to demonstrate significant activity in GC.

Overlapping function with other nuclear hormone receptors is well known as well and signaling in these pathways is not addressed. Furthermore, all experiments were carried out in male nude mice. No female mice were used.

Response: As the reviewer points out, different nuclear receptors (NRs) do have overlapping functions and some have roles in the Wnt signaling pathway. For example, PPAR gamma and delta influence Wnt signaling. Thus, there was a possibility ESRRG might interact with PPARs to regulate Wnt signaling. We now address this in our revised text (Page 13, line 18). Whereas nuclear receptors (NRs) play a crucial role in hormone-related cancers such as breast and prostate cancer, their role of NR is not well described in GC. In our current study, we have demonstrated the interplay between ESRRG and Wnt signaling in GC for the first time.

As the reviewer also mentions, we previously used only male mice. Many researchers in fact do so to avoid the potentially complicating effects associated with a changing female hormonal environment. In our revised experiments however, we used female mice in our *in vivo* analysis. As shown in Figure 4e, the anti-tumor effects ESRRG are not sex-dependent.

Reviewer #3 (Remarks to the Author):

Kang et al. describe a tumor suppressive effect of the nuclear receptor ERRgamma in gastric cancer. The data clearly shows decreased expression in tumor relative to non-tumor samples, and decreased expression is correlated with poorer clinical outcomes. Conversely ERRgamma overexpression or agonist activation decreases growth of gastric cancer cell lines and tumor xenografts. Anti-proliferative effects of ERRgamma are linked to inhibition of beta-catenin activity.

The primary concern is that the basis for the antagonism of beta-catenin by ERRgamma is quite unclear. It is suggested that beta-catenin phosphorylation is altered, but the data supporting this is not compelling. It is also suggested that binding of overexpressed LEF to a site in the cyclin D1 promoter is decreased by ERRgamma overexpression, but the basis for this is unclear, and this experiment should have been done with endogenous LEF. ERRgamma overexpression also strongly decreases transactivation of the Top Flash reporter, but the basis for this effect is not clear. Overall, the impact of ERRgamma on Wnt signaling is clear, but the lack of a clear mechanism to account for this leaves the story incomplete.

Response: As suggested, we have added new molecular data to our revised manuscript to support our hypothesis that ESRRG suppresses Wnt signaling. As the reviewer points out, our previous evidence that β -catenin phosphorylation is altered by ESRRG was not particularly compelling. Phosphorylated β -catenin is an indicator of β -catenin activity. Notably however, once β -catenin is phosphorylated, it is degraded in the cytoplasm and, as a transcription factor, ESRRG is principally expressed in the nucleus. As shown in Figure 5a however, ESRRG enhances phosphorylated β -catenin activity. Since the cellular localization of ESRRG and

phosphorylated β -catenin differ, we speculate that ESRRG might not directly influence β -catenin activity and that the increases in phosphorylated β -catenin might be due to the suppression of cell growth by ESRRG. Thus, we investigated how the ESRRG transcription factor antagonizes TCF4/LEF1 on the *CCND1* promoter. We show from our analysis using IP and FCCS assays (see responses above) that ESRRG directly interacts with TCF4/LEF1 (Fig. 5g-i). In addition, ChIP analysis revealed that endogenous TCF4/LEF1 binding to the *CCND1* promoter was inhibited by ESRRG in GC cells (Fig. 5e-f). When we performed ChIP using ESRRG antibodies, *CCND1* promoter-bound TCF4/LEF1 was recruited by ESRRG. In addition, the transcriptional activity of TCF4/LEF was inhibited by ESRRG (Fig. 4h) and the gene expression of TCF4/LEF1 in nucleus was suppressed by ESRRG (Fig. 5b). We suggest from these data that ESRRG directly interacts with TCF4/LEF1 bound on the *CCND1* promoter and thereby antagonizes Wnt signaling (Fig 5 and 7).

To demonstrate that ESRRG overexpression strongly decreases the transactivation of β -catenin, we performed reporter assays using the Top Flash reporter as it contains a binding site for β -catenin. The reviewer mentioned that the basis for this effect was not clear previously. Our new data indicated that in the nucleus, activated Wnt signaling enhances the transcriptional activity of β -catenin and the TCF/LEF complex. We then found that ESRRG overexpression strongly decreases the transactivation of β -catenin (Fig. 4g). In addition, we also performed a reporter assay using the *CCND1* promoter, TCF4/LEF1 and ESRRG. Enhanced transactivation of *CCND1* by TCF4/LEF1 was significantly suppressed by ESRRG (Fig. 4h). Our data thus clearly show that ESRRG suppresses the transcriptional activity of Wnt-signaling via TCF/LEF1 binding.

In conclusion, our study clearly demonstrates that ESRRG antagonizes the Wnt components TCF4/LEF1 through a physical interaction as a transcriptional suppressor in GC cells.

Reviewers' comments:

Reviewer #1 (Remarks to the Author):

This reviewer criticized that the lack of the mechanistic analyses of the inhibitory activity of ESRRG for Wnt signaling in the original manuscript. The authors performed several experiments and added new results in the revised manuscript. However, it is still hard to understand how ESRRG inhibits Wnt signaling.

1. The authors showed that ESRRG stimulates Ser33/37/Thr41 phosphorylation of β -catenin in the cytoplasm and promotes the degradation of β -catenin in the nucleus (Fig. 5b-d). The data in Fig. 5b is not convincing. How were protein expression levels normalized? The HDAC1 expression level also seems to be reduced in the nucleus by the expression of FLAG-ESRRG. In addition, is ubiquitination involved in ESRRG-induced degradation of β -catenin? An important question is how ESRRG promotes the ubiquitination and phosphorylation of β -catenin in the authors' model. The authors should know that β -catenin is phosphorylated by GSK-3 and casein kinase I in the axin complex including APC.

2. The authors also said that TCF4 and LEF1 are degraded by the expression of FLAG-ESRRG. What is the mechanism? The experiment in Fig. 5c is weird. By the treatment with CHX, the expression of FLAG-ESRRG was also decreased. The authors should carefully evaluate the results under the conditions.

3. The raw data (real DNA bands in the gel) should be shown in Fig. 5e and f.

4. The authors demonstrated that ESRRG co-localizes with TCF4/LEF1 in the nucleus using FCCS (Fig. 5h-j). A negative control would be necessary for this kind of assay. Since the authors showed that ESRRG does not bind to β -catenin in Fig. 5g, β -catenin must be a good negative control.

5. If the authors believe that DY131 inhibits Wnt signaling by ESRRG expression, the effect of ESRRG knockdown on DY131-induced phenotypes must be examined.

6. The inhibitory effect of ESRRG on β -catenin-, TCF4-, and LEF1-dependent transcription was examined using the reporter gene assay in Fig. 4g and h. Does endogenous Axin2 mRNA level decrease in this experiment?

7. The rescue effects on tumor growth by re-expression of TCF4/LEF1 in Figs. 4e and 6b are too small.

Reviewer #2 (Remarks to the Author):

The authors responded adequately to most queries.

Reviewer #3 (Remarks to the Author):

The additional data showing strong suppression of LEF1/TCF4 binding by ESRRG overexpression in 2 cell types still leaves questions, but adequately addresses prior concerns on basic mechanism.

Reviewers' comments:

Reviewer #1 (Remarks to the Author):

This reviewer criticized that the lack of the mechanistic analyses of the inhibitory activity of ESRRG for Wnt signaling in the original manuscript. The authors performed several experiments and added new results in the revised manuscript. However, it is still hard to understand how ESRRG inhibits Wnt signaling.

Response: This reviewer has raised some mechanistic issues which are important to clarify. As we mention in the manuscript, our principal proposal in this regard is that ESRRG antagonizes Wnt-signaling by suppressing the transcriptional activity of TCF4/LEF1 through a direct interaction. This leads to a block in target DNA binding by TCF4/LEF1 and hence tumor suppression. Although ESRRG alters β -catenin levels in the nucleus and β -catenin phosphorylation levels in the cytoplasm, it does not physically interact with β -catenin and is not localized in cytoplasm. This suggests that these are indirect effects of ESRRG due to decreased cell growth. ESRRG plays a role as a transcription factor that modulates gene regulation. Hence, the major biological events related to ESRRG occur in the nucleus. We speculate that ESRRG functions as a transcriptional repressor of TCF4/LEF1, which are major transcriptional components of the Wnt-signaling pathway (Figure 7). We hope that this description provides some clarity to the reviewer. We have depicted this proposed mechanism of transcriptional regulation by ESRRG in the schematic figure below.

Figure: Regulation of Wnt-signaling genes by ESRRG

1. The authors showed that ESRRG stimulates Ser33/37/Thr41 phosphorylation of β-catenin in the cytoplasm and promotes the degradation of β-catenin in the nucleus (Fig. 5b-d). The data in Fig. 5b is not convincing. How were protein expression levels normalized? The HDAC1 expression level also seems to be reduced in the nucleus by the expression of FLAG-ESRRG. In addition, is ubiquitination involved in ESRRG-induced degradation of β-catenin? An important question is how ESRRG promotes the ubiquitination and phosphorylation of β-catenin in the

authors' model. The authors should know that β -catenin is phosphorylated by GSK-3 and casein kinase I in the axin complex including APC.

Response: As shown in Figure 5b, the nuclear fraction was normalized with HDAC1 protein levels. The reviewer mentions that HDAC1 was decreased by Flag-ESRRG but this was very marginal compared to the reduction in β -catenin, TCF4 and LEF1 by Flag ESRRG. Thus, we can conclude that ESRRG suppresses β -catenin in the nucleus where ESRRG is principally expressed. ESRRG directly interacts with TCF4/LEF1 (the major transcription factor in the Wnt-signaling pathway), which forms a complex with β -catenin in the nucleus. As ESRRG suppresses TCF4/LEF1, β -catenin expression in the nucleus is also indirectly inhibited by ESRRG.

The reviewer queried whether ubiquitination is involved in the ESRRG-induced degradation of β -catenin. To assess this, we treated AGS cells with the MG132 proteasome inhibitor after ESRRG infection. In the ESRRG-infected cells, β -catenin phosphorylation was increased which is consistent with our previous results. MG132 treatment led to β -catenin phosphorylation. However, β -catenin phosphorylation was not markedly increased in ESRRG-infected cells treated with MG132 when compared with MG132 treated control cells or untreated ESRRG-infected cells. (Supplementary Figure 10b). In addition, the total and phospho-GSK3 α/β expression levels were not altered by ESRRG. Most importantly, MG132 did not markedly alter β -catenin protein levels in ESRRG expressing cells. These data indicate that ubiquitination is not a major contributor to ESRRG-induced degradation of β -catenin and that degradation of β -catenin induced by ESRRG expression is not dependent on activation of GSK3 α/β , which phosphorylates β -catenin.

The reviewer has also raised concerns about how ESRRG influences β -catenin phosphorylation in the cytoplasm. ESRRG is a nuclear receptor functioning as a transcription factor and is thus primarily expressed in the nucleus, as confirmed in our own experiments (Fig. 5h-i). Thus, increased β -catenin phosphorylation by ESRRG in the cytoplasm is likely an indirect effect by an as yet unknown mechanism potentially a consequence of suppression of cell growth by ESRRG (please see the schematic Figure above). Usually, phosphorylation and ubiquitination events are mediated by direct interaction with the counter partner. However, β -catenin does not directly interact with ESRRG. As we show clearly in Figures 4g-h and Figure 5g-i, ESRRG functions as a transcriptional repressor of TCF4/LEF1 via a direct interaction. Although we demonstrated from our current data that some ESRRG driven events such as β -catenin phosphorylation occur in the cytoplasm, these are not primary functions of ESRRG. Transcriptional suppression of TCF4/LEF1 appears to be the primary mechanism by which ESRRG antagonizes Wnt-signaling. We describe this in the manuscript on page 12.

2. The authors also said that TCF4 and LEF1 are degraded by the expression of FLAG-ESRRG. What is the mechanism? The experiment in Fig. 5c is weird. By the treatment with CHX, the expression of FLAG-ESRRG was also decreased. The authors should carefully evaluate the results under the conditions.

Response: β -catenin forms a complex with TCF4 and LEF1 that can lead to cancer cell proliferation and the stability of those proteins is crucial for the maintenance of their gene functions (Ref. Curr Opin Cell Biol. 1999 Apr; 11(2):233-40. Regulation of LEF1/TCF4 transcription factors by Wnt and other signals). Thus, we measured protein stability of ESRRG in the presence of CHX. As the reviewer mentions, Flag-ESRRG expression was decreased by CHX but ESRRG was still expressed and its activity was sustained. Since

ESRRG directly interacts with TCF4/LEF1 but not β -catenin, and functions as transcriptional repressor, it may influence LEF1/TCF4 protein stability by regulating gene transcription.

3. The raw data (real DNA bands in the gel) should be shown in Fig. 5e and f.

Response: As indicated in Figures 5e and f, we performed CHIP assays to measure protein/DNA binding which we quantified using real time PCR. This real time system is the most reliable way to conduct these assays. As the reviewer mentions, semi-quantitative PCR has been typically used to visualize the DNA bands in the gel. But this method is not completely precise as the total cycle numbers can be adjusted. The band density of the PCR product can thus be altered by varying the cycle numbers regardless of the actual level of direct protein and DNA binding. Thus, we used the qRT-PCR method. The raw Ct data from the qRT-PCR are provided for the reviewer (Review's only data).

4. The authors demonstrated that ESRRG co-localizes with TCF4/LEF11 in the nucleus using FCCS (Fig. 5h-j). A negative control would be necessary for this kind of assay. Since the authors showed that ESRRG does not bind to β -catenin in Fig. 5g, β -catenin must be a good negative control.

Response: Accordingly, we have added negative control (β -catenin) data (Supplementary Figure 10c and d). In the FCCS analysis, ESRRG did not interact with β -catenin.

5. If the authors believe that DY131 inhibits Wnt signaling by ESRRG expression, the effect of ESRRG knockdown on DY131-induced phenotypes must be examined.

Response: We propose that DY131 acts through ESRRG. Accordingly, we carried out studies as requested by the reviewer to show DY131 is an ESRRG agonist. Whereas DY131 inhibited cell growth in control cells, it did not inhibit proliferation in siESRRG-treated cells.

(Supplementary Figure 12). These data demonstrate that the anti-proliferative effects of DY131 are mediated via ESRRG-demonstrating that DY131 is ESRRG agonist.

6. The inhibitory effect of ESRRG on β -catenin-, TCF4-, and LEF11-dependent transcription was examined using the reporter gene assay in Fig. 4g and h. Does endogenous Axin2 mRNA level decrease in this experiment?

Response: The reporter assay monitors gene transcriptional activity and not mRNA levels. We had already shown that the Axin2 level was significantly decreased by ESRRG in microarray and qRT-PCR experiments (Fig. 4a and b).

7. The rescue effects on tumor growth by re-expression of TCF4/LEF11 in Figs. 4e and 6b are too small.

Response: In our current study, we demonstrate that ESRRG functions as a tumor suppressor by antagonizing the Wnt-signaling pathway. Although we suggest from our data that ESRRG targets Wnt-signaling, it may also inhibit diverse oncogenic signaling pathways that have yet to be unidentified. Thus, although we rescued TCF4/LEF1 after ESRRG-overexpression, the suppressed cell or tumor growth did not return to the control levels since we only rescued LEF1/TCF4 and other factors may be involved in the cell growth suppression. The reviewer mentions that the rescue of tumor and cell growth was small but all of our rescue experiment results were significant. Furthermore, tumor weight was increased following the re-introduction of TCF4/LEF1 in ESRRG overexpression cells, demonstrating that Wnt-signaling is the principal pathway by which ESRRG suppresses gastric cancer cell growth in in vivo models.

REVIEWERS' COMMENTS:

Reviewer #1 (Remarks to the Author):

This reviewer agrees that the authors experimentally proved the inhibition of TCF4/LEF1 transcription activity by the binding of ESRRG to them. However, as the authors mentioned, ESRRG reduced the expression levels of β -catenin and TCF4/LEF1 in the nucleus by an unknown mechanism. This mechanism may be more important in the tumor suppressor activity of ESRRG.

REVIEWERS' COMMENTS:

Reviewer #1 (Remarks to the Author):

This reviewer agrees that the authors experimentally proved the inhibition of TCF4/LEF1 transcription activity by the binding of ESRRG to them. However, as the authors mentioned, ESRRG reduced the expression levels of β -catenin and TCF4/LEF1 in the nucleus by an unknown mechanism. This mechanism may be more important in the tumor suppressor activity of ESRRG.

Response: Regulation of TCF4/LEF1 is directly controlled by binding to and activating a consensus LEF4/LEF1 binding site within its own promoter-suggesting that TCF4/LEF1 activates its own gene expression in nucleus. ESRRG inhibits transcription activity of TCF4/LEF1 and finally suppresses gene expression level of TCF4/LEF1. Since CTNNB1 activity is influenced by TCF4/LEF1, CTNNB1 expression is also inhibited by ESRRG. We described in "Discussion" part (Page 12).